# The virulence regulator VirB from *Shigella flexneri* uses a CTP-dependent switch mechanism to activate gene expression

Sara Jakob[1], Wieland Steinchen[2,3], Juri Hanßmann[1,4], Julia Rosum[1], Katja Langenfeld[5], Manuel Osorio-Valeriano[1,9], Niklas Steube[6], Pietro I. Giammarinaro[2,10], Georg K. A. Hochberg[2,3,6], Timo Glatter[7], Gert Bange[2,3,8], Andreas Diepold[5] & Martin Thanbichler[1,3,4] ✉

The transcriptional antisilencer VirB acts as a master regulator of virulence gene expression in the human pathogen *Shigella flexneri*. It binds DNA sequences (*virS*) upstream of VirB-dependent promoters and counteracts their silencing by the nucleoid-organizing protein H-NS. However, its precise mode of action remains unclear. Notably, VirB is not a classical transcription factor but related to ParB-type DNA-partitioning proteins, which have recently been recognized as DNA-sliding clamps using CTP binding and hydrolysis to control their DNA entry gate. Here, we show that VirB binds CTP, embraces DNA in a clamp-like fashion upon its CTP-dependent loading at *virS* sites and slides laterally on DNA after clamp closure. Mutations that prevent CTP-binding block VirB loading in vitro and abolish the formation of VirB nucleoprotein complexes as well as virulence gene expression in vivo. Thus, VirB represents a CTP-dependent molecular switch that uses a loading-and-sliding mechanism to control transcription during bacterial pathogenesis.

*Shigella* species are the causative agents of bacillary dysentery[1] and the second leading cause of diarrheal mortality, with more than 200,000 deaths per year worldwide[2]. After ingestion, they invade the colonial epithelium, replicate in the cytoplasm of epithelial cells, and then propagate the infection by spreading from cell to cell, driven by actin-based motility. These processes are facilitated by a multitude of virulence proteins, known as effectors, which are produced in two consecutive waves and directly injected into the host cytoplasm by means of a type III secretion system (T3SS)[3]. The T3SS apparatus as well as most effectors are encoded on a large (~220 kb) virulence plasmid, named pINV[4–6]. Many of the pINV-associated genes are

controlled at the transcriptional levels by a three-tiered regulatory cascade[7] that is triggered upon transition of *Shigella* cells to human body temperature (37 °C)[8]. It initiates with the production of the transcriptional activator VirF[9], driven by temperature-induced changes in the structure of the *virF* promoter region[10–14]. Among the regulatory targets of VirF is the gene for the second-tier regulator VirB[15–17], which activates the expression of ~50 genes, most of which are organized into two large operons coding for structural components of the T3SS and effectors mediating host invasion[18]. The VirB regulon also includes the gene for the third-tier regulator MxiE[18], which later promotes the production of effectors involved in

[1]Department of Biology, University of Marburg, Marburg, Germany. [2]Department of Chemistry, University of Marburg, Marburg, Germany. [3]Center for Synthetic Microbiology (SYNMIKRO), Marburg, Germany. [4]Max Planck Fellow Group Bacterial Cell Biology, Max Planck Institute for Terrestrial Microbiology, Marburg, Germany. [5]Department of Ecophysiology, Max Planck Institute for Terrestrial Microbiology, Marburg, Germany. [6]Evolutionary Biochemistry Group, Max Planck Institute for Terrestrial Microbiology, Marburg, Germany. [7]Mass Spectrometry and Proteomics Facility, Max Planck Institute for Terrestrial Microbiology, Marburg, Germany. [8]Max Planck Fellow Group Molecular Physiology of Microbes, Max Planck Institute for Terrestrial Microbiology, Marburg, Germany. [9]Present address: Department of Cell Biology, Blavatnik Institute, Harvard Medical School, Boston, MA, USA. [10]Present address: Heidelberg University Biochemistry Center (BZH), Heidelberg, Germany. ✉e-mail: thanbichler@uni-marburg.de

post-invasion processes, including the subversion of host immune response, innate immunity, and cell death pathways[1,7].

Due to its central role in virulence gene expression, VirB is critical for pathogenicity in *Shigella* and mutants lacking this regulator are avirulent[17,19,20]. Interestingly, while VirF and MxiE are classical AraC-type transcriptional activators[9,21,22], VirB is an atypical transcription factor related to the ParB family of DNA-partitioning proteins[23–26]. Members of this family usually function in the context of ParAB*S* DNA segregation systems, where they serve as centromere-binding proteins that interact dynamically with the partition ATPase ParA to mediate the separation of newly replicated sister chromosomes or low-copy number plasmids[27]. ParB proteins form dimers and interact specifically with short (16 bp) palindromic DNA sequences called *parS* that are clustered close to the replication origin of target molecules[28,29]. After initial specific binding to individual *parS* sites, they spread laterally into the neighboring DNA segments, forming a large nucleoprotein assembly (partition complex) that typically includes 10–20 kb of the origin region[30–33]. Recent work has clarified the mechanistic basis of the spreading process by showing that ParB proteins constitute a novel class of molecular switches and act as DNA-sliding clamps that use cytidine triphosphate (CTP) binding and hydrolysis to control the opening state of their DNA entry gate[34–39]. The CTP-binding site is located in the N-terminal ParB/Sulfiredoxin (Srx) domain of ParB[38,39], which is followed by a central helix-turn-helix (HTH) *parS*-binding domain[40–43], a non-structured linker region and a C-terminal dimerization domain, which tightly links the two subunits of a ParB dimer[41,44]. Upon juxtaposition of the HTH domains on a palindromic *parS* site, the two N-terminal ParB/Srx domains associate in a CTP-dependent manner and thus close the ParB dimer into a ring-like structure, with bound DNA entrapped in between the two subunits[35,37,39]. Ring closure releases the HTH domains from *parS* and thus enables the ParB clamp to slide into the flanking DNA regions, making *parS* accessible again to other ParB dimers[35,39,45,46]. The slow intrinsic CTPase activity of ParB rings eventually leads to CTP hydrolysis, triggering clamp opening and the dissociation of ParB from the DNA[34,36,37]. Spontaneous nucleotide exchange then initiates the next loading cycle. This sequence of events leads to the establishment of a 1D diffusion gradient of ParB clamps within the origin region, originating at *parS* clusters and shaped by the CTPase activity of ParB.

Intriguingly, the regulatory activity of VirB depends on its interaction with *parS*-like palindromic sequences (henceforth called *virS*) in the target promoter regions[23,25,26,47–49]. These recognition sites can be located more than 1 kb upstream of the corresponding transcriptional start sites[48,50], indicating that VirB does not stimulate gene expression through a direct interaction with RNA polymerase. A series of studies showed that VirB indeed acts at a distance and counteracts the silencing of target promoters by the histone-like nucleoid-structuring protein H-NS, which associates with AT-rich regions in the surroundings or downstream of *virS* sites[26,48–52]. This anti-silencing mechanism was proposed to involve the spreading of VirB from its DNA recognition site, because the insertion of a roadblock between *virS* and the H-NS binding regions abolished the regulatory effect of VirB[49]. Consistent with this notion, recent work has shown that the *virS*-dependent interaction of VirB with promoter regions causes large-scale changes in DNA topology, leading to a local decrease in negative supercoiling that alleviates H-NS-mediated gene silencing[53]. However, the molecular mechanism underlying the spreading of VirB and its effect on DNA topology are incompletely understood.

The homology of VirB to ParB proteins, its dependence on a *parS*-like binding site, and its potential spreading within promoter regions raise the possibility that VirB could use a clamping-and-sliding mechanism similar to that reported for ParB to associate with its target DNA. Here, we use a combination of structural modeling, molecular interaction studies, and hydrogen-deuterium exchange mass spectrometry combined with in vivo localization and functional studies to dissect the mode of action of VirB. We demonstrate that VirB binds CTP but appears to lack appreciable CTPase activity. Consistent with previous results[25,54], we show that VirB has strong non-specific DNA-binding activity in standard buffers. However, under more stringent conditions, it accumulates on DNA in a strictly *virS* and CTP-dependent manner, with *virS* acting as an entry site that mediates the loading of multiple VirB dimers, which diffuse away from *virS* after the loading step. We further observe that the two N-terminal nucleotide-binding domains of VirB homodimerize in the presence of CTP, in a process stimulated by *virS* DNA. Finally, we show that the CTP-dependent closure of VirB clamps is a prerequisite for the formation of VirB nucleoprotein complexes and the induction of virulence gene expression in *S. flexneri*. Together, these results indicate that VirB constitutes a distinct class of CTP-dependent molecular switches that uses a loading-and-sliding mechanism control gene expression during bacterial pathogenesis.

## Results

### VirB is a homolog of plasmid-encoded ParB proteins with CTP-binding activity

Previous work has revealed a significant similarity of VirB to plasmid-encoded ParB proteins in terms of its amino acid sequence and domain architecture as well as its ability to interact specifically with a *parS*-like sequence motif[23,24,26]. Moreover, crystallographic studies showed that the *virS*-binding domain of VirB is structurally related to the HTH domains of the ParB homologs ParB, SopB, and KorB, encoded by the *E. coli* plasmids P1, F, and RP4, respectively[25]. Prompted by these findings, we aimed to investigate whether VirB also shared the ability of ParB proteins to bind and hydrolyze CTP. To this end, we first generated an amino acid sequence alignment comparing the N-terminal domain of VirB with the corresponding N-terminal nucleotide-binding domain of various well-characterized ParB homologs from plasmid or chromosome partitioning systems. Importantly, three regions that critically contribute to CTP binding in ParB are also conserved in VirB (Fig. 1a), including the C-motif, the GxRR motif, and the P-loop, which coordinate the cytidine base, the triphosphate moiety, and the γ-phosphate group of the CTP nucleotide, respectively[36,37,39]. However, VirB features only one out of the two conserved residues required for CTP hydrolysis in chromosomally encoded ParB proteins[34,36,37] (Fig. 1a), suggesting that it may be able to bind CTP but have no or strongly reduced CTPase activity.

To further analyze the function of VirB, we generated a structural model of a VirB homodimer, using AlphaFold-Multimer[55] (Fig. 1b). The model obtained suggests that VirB can form a clamp-like structure similar to that reported for ParB[36,37,39], with the two subunits interacting in the N-terminal and C-terminal regions. Consistent with this prediction, mass photometry data show that VirB exists as a dimer in solution (Supplementary Fig. 1). The center of the closed dimeric complex features an opening flanked by non-structured linker regions that contains an abundance of positively charged residues and is large enough to accommodate a DNA molecule (Supplementary Fig. 2). As in ParB, the interaction in the N-terminal regions is predicted to rely on the homodimerization of the two ParB/Srx nucleotide-binding domains (NBDs), which involves a crossover of the two polypeptide chains that places the NBD of each *cis*-subunit next to the *virS*-binding domain (VBD) of the *trans*-subunit. A superimposition of the structural model of VirB with the crystal structure of the chromosomally encoded ParB homolog from *Myxococcus xanthus*[37] indicates that the NBDs of the two proteins have a similar fold, especially in regions forming the nucleotide-binding pocket of ParB, with a root-mean-square deviation (RMSD) of 1.51 Å for 45 paired $C_\alpha$ atoms (Fig. 1c). However, VirB is distinguished from ParB by an N-terminal helix as well as two adjacent long antiparallel β-strands that appear to connect the

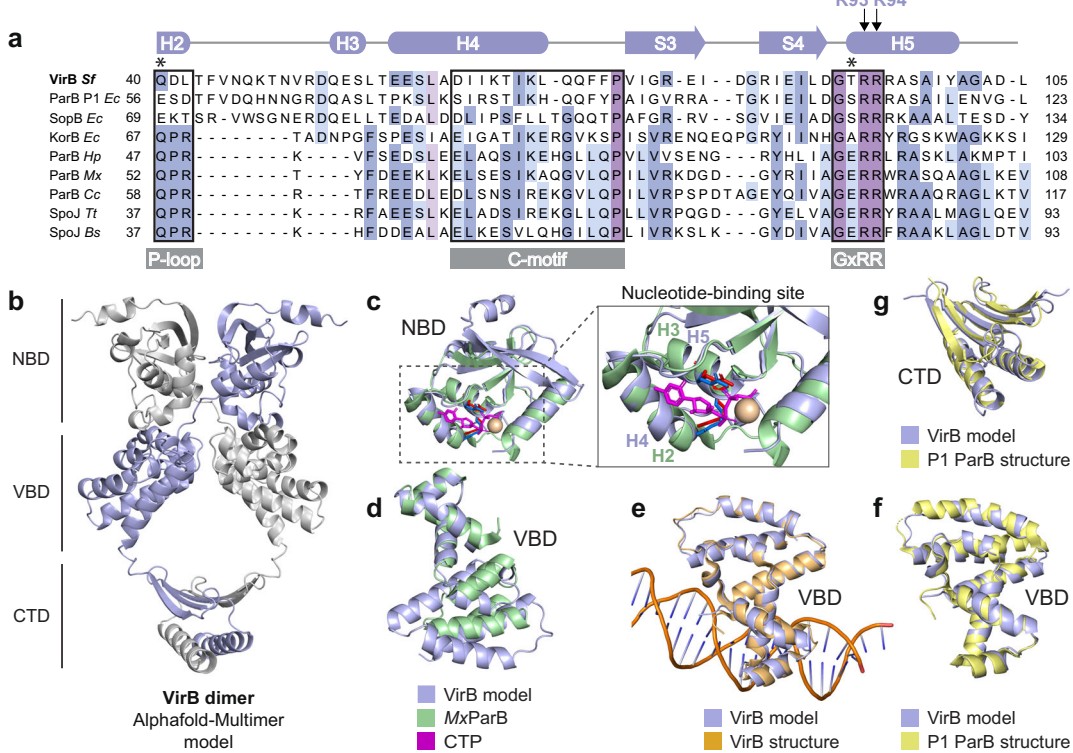

**Fig. 1 | VirB is structurally related to plasmid-encoded ParB proteins. a** Multiple sequence alignment comparing VirB from *S. flexneri* with plasmid- and chromosomally encoded ParB orthologs, with a focus on the nucleotide-binding region. Conserved motifs with critical roles in nucleotide binding, including the P-loop, the C-motif, and GxRR motif[37], are highlighted by black frames. The schematic at the top represents the predicted secondary structure of VirB. Two conserved arginine residues (R93 and R94) in the GxRR motif, which were reported to be essential for nucleotide binding in other family members, are marked. Asterisk indicate the residues that were shown to be critical for CTP hydrolysis in chromosomally encoded ParB proteins[34,36,37]. The proteins aligned and their UniProt accession numbers are VirB of *S. flexneri* (P0A247), ParB from *Escherichia coli* plasmid P1 (P07621), SopB from *E. coli* plasmid F (P62558); KorB from *E. coli* plasmid RK2 (P07674), ParB from *Helicobacter pylori* (O25758), ParB from *Myxococcus xanthus* (Q1CVJ4), ParB from *Caulobacter crescentus* (P0CAV8), SpoJ from *Thermos thermophilus* (Q72H91) and SpoJ from *Bacillus subtilis* (P26497). **b** Structural model of an *S. flexneri* VirB dimer, generated with AlphaFold-Multimer[55]. The nucleotide-binding domain (NBD), the *virS*-binding domain (VBD), and the C-terminal dimerization domain (CTD) are indicated. Interface pTM (ipTM) score: 0.776. **c** Overlay of the predicted structure of the NBD of VirB with the crystal structure of the NBD of *M. xanthus* ParB-Q52A bound to CTPγS (*Mx*ParB) (PDB: 7BNR), with an RMSD of 1.512 Å for 45 aligned $C_\alpha$ atoms. The close-up shows the nucleotide-binding site, with the conserved arginine residues in the GxRR motif of VirB (R93 and R94 in helix H5) shown in blue and the equivalent residues (R94 and R95 in helix H3) of *Mx*ParB shown in red. **d** Overlay of the predicted structure of the VBD of VirB with the crystal structure of the VBD of *M. xanthus* ParB-Q52A bound to CTPγS (*Mx*ParB) (PDB: 7BNR). **e** Overlay of the modeled VBD of VirB with the crystal structure of the VBD bound to the *virS* site of the *S. flexneri icsB* gene (PDB: 3W3C), with an RMSD of 0.883 Å for 98 aligned $C_\alpha$ atoms. **f** Overlay of the modeled VBD of VirB with the crystal structure of the DNA-bound HTH-domain of *E. coli* plasmid P1 ParB (PDB: 1ZX4), with an RMSD of 0.811 Å for 93 aligned $C_\alpha$ atoms. **g** Overlay of the CTD of modeled VirB with the corresponding domain of *E. coli* plasmid P1 ParB (PDB: 1ZX4), with an RMSD of 1.274 Å for 90 aligned $C_\alpha$ atoms.

nucleotide-binding region of VirB to an opposing α-helix (corresponding to helix 4 of ParB), thereby potentially reducing the conformational flexibility of the NBD. The predicted structure of the VBD shows differences to the corresponding *parS*-binding domain of *M. xanthus* ParB (Fig. 1d) but is highly similar to a previously solved crystal structure of the VBD in complex with *virS* DNA[25] (RMSD of 0.883 Å for 98 paired $C_\alpha$ atoms), underscoring the validity the modeling approach (Fig. 1e). Notably, both the VBD and CTD of VirB have striking structural similarity with the respective domains of ParB from *E. coli* plasmid P1 (Fig. 1f, g), supporting the notion that VirB may have evolved from plasmid partitioning proteins.

**VirB has CTP-binding activity and shows weak CTPase activity**

Since key features of the nucleotide-binding pocket of ParB were conserved in VirB, we aimed to investigate whether VirB was also able to interact with CTP, using quantitative nucleotide-binding assays based on isothermal titration calorimetry (ITC). First, we analyzed the binding of the poorly hydrolyzable CTP analog CTPγS, which was chosen to avoid potential adverse effects of CTP hydrolysis on the measurements. We observed that VirB bound this nucleotide with an affinity ($K_D \approx 12\,\mu M$) that is comparable to the one measured for

ParB[34,37] and high enough to ensure its saturation under in vivo conditions[56] (Fig. 2a and Supplementary Fig. 3a). We also observed an interaction of VirB with cytidine diphosphate (CDP), although the binding affinity for this nucleotide was more than 7-fold lower ($K_D \approx 91\,\mu M$) (Fig. 2b and Supplementary Fig. 3b). Similar results were obtained using microscale thermophoresis as an alternative technique to study the nucleotide-binding behavior of VirB (Fig. 2c and Supplementary Fig. 3c). Notably, CTP binding was drastically reduced upon the mutation of arginine residues (R93, R94) in the conserved GxRR motif to alanines (see Fig. 1a), suggesting that VirB and ParB share a similar mode of nucleotide binding. Given that some proteins, such as sulfiredoxins[57,58] and free-serine kinases[59,60], contain ParB/Srx-like domains with ATP-binding activity, we also tested VirB for its ability to interact with ATP. However, no significant binding was observed (Fig. 2c and Supplementary Fig. 3c).

We then went on to clarify whether VirB was able to hydrolyze the bound CTP. In the case of ParB, significant CTPase activity is only detectable when the protein is incubated with both nucleotide and *parS* DNA. Similarly, the basal CTPase activity of VirB was very low in the absence of *virS* DNA and increased twofold in the presence of its binding target (Fig. 2d). However, the maximal turnover number

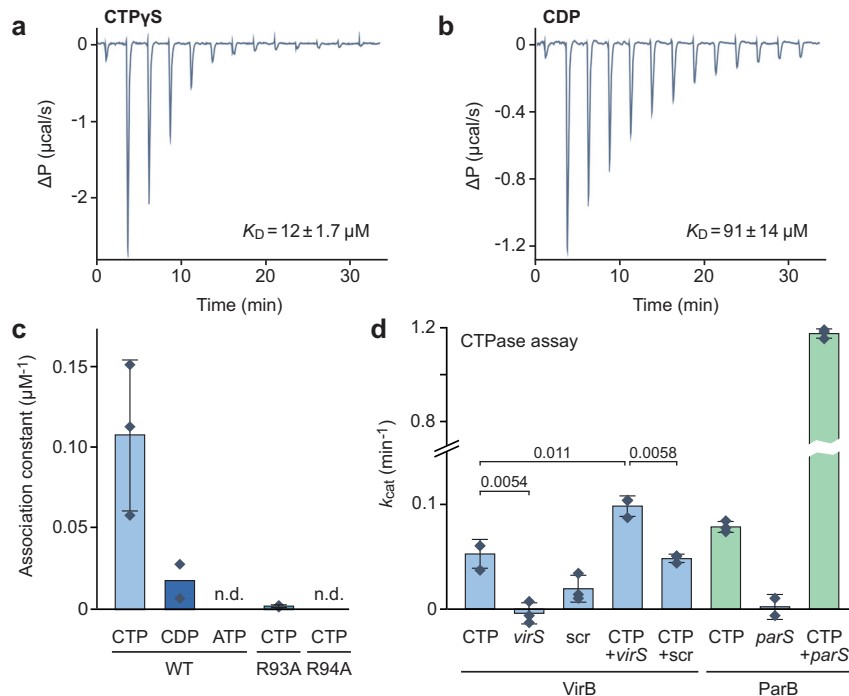

**Fig. 2 | VirB is a CTP-binding protein. a, b** Isothermal titration calorimetry analysis of the interaction of VirB with **a** CTPγS and **b** CDP. A solution of VirB (115 μM) was titrated with a stock solution (1.55 mM) of the indicated nucleotides. The graphs indicate the heat changes observed after each of the 13 injections. The $K_D$ values obtained are given in the graphs. **c** Microscale thermophoresis analysis of the interaction of VirB (WT) and its R93A and R94A variants with CTP, CDP and ATP. Bars show the mean equilibrium association constants ($1/K_D$) obtained for the indicated conditions (±SD, when indicated). Data points (diamonds) represent the results of $n = 3$ (CTP binding to WT and R93A) or 2 (all other reactions) independent experiments, each of which was performed in triplicate. The underlying titration curves are given in Supplementary Fig. 3c. The corresponding $K_D$ values are: WT-

CTP (11 μM), WT-CDP (88 μM), and R93A-CTP (1.3 mM). n.d. not detectable.
**d** CTPase activities of VirB and *Mx*ParB. The indicated proteins (5 μM) were incubated with 1 mM CTP and/or double-stranded DNA oligonucleotides containing a *virS* site (0.5 μM), a scrambled *virS* site (0.5 μM) or an *M. xanthus parS* site (0.3 μM). CTP hydrolysis rates were determined using an NADH-coupled enzyme assay. The values obtained were corrected for the background activity measured in reactions containing only the respective protein but no ligands. Columns represent the mean (±SD) of three independent experiments (diamonds), each of which was performed in triplicate. The statistical significance of differences between reactions was determined using an unpaired two-sided Welch's *t*-test. Relevant *p* values are given in the graph. Source data are provided as a Source Data file.

reached ($k_{cat} \approx 0.1 \text{ min}^{-1}$) was still considerably lower than that obtained for *M. xanthus* ParB, although it is possible that additional, thus-far unknown factors are required to further stimulate the hydrolytic reaction.

**VirB is loaded onto DNA in a CTP- and *virS*-dependent manner**

VirB was shown to specifically interact with *virS* in vitro and to require the presence of virS upstream of target promoters for its regulatory activity in vivo[25,26,48,49,51,53]. To clarify the role of CTP binding in the interaction of VirB with target DNA regions, we analyzed its DNA-binding behavior using a previously established biolayer interferometry assay[35]. For this purpose, we amplified the *icsB−ipgD* intergenic region of pINV with its centrally located *virS* site, which mediates the VirB-dependent regulation of two divergent large virulence operons in the so-called Entry Region of *S. flexneri*[26,61] (Supplementary Fig. 4). The resulting DNA fragment (215 bp) was then immobilized on biosensors such that both of their ends were stably linked to the sensor surface (Fig. 3a) and probed with VirB in the absence or presence of CTP. In both conditions, VirB showed strong DNA-binding activity. However, nucleotide-free reactions reproducibly yielded biphasic association curves when VirB was used at elevated concentrations, with the signal decreasing markedly during the loading phase (Fig. 3b). In the presence of CTP, by contrast, VirB stably associated with the DNA in all cases (Fig. 3c). However, almost identical results were obtained with closed DNA fragments lacking a *virS* sequence (Supplementary Fig. 5). Similarly, VirB also showed a strong *virS*-independent interaction with biosensors carrying open double-stranded oligonucleotides, although the presence of CTP again

appeared to modulate its binding behavior (Supplementary Fig. 7). These findings imply that VirB has strong non-specific DNA-binding activity, consistent with the high density of positive charges in its VBD and CTD (Supplementary Fig. 2a). Under the conditions used (150 mM NaCl), this property may obscure the specific, *virS*-mediated binding of VirB to DNA as well as potential effects of CTP on this interaction in vitro.

It was conceivable that more stringent conditions were required to discriminate between specific and non-specific interactions of VirB with its target DNA. We therefore repeated the binding assays at elevated salt concentrations (500 mM NaCl) to weaken electrostatic interactions between positively charged residues and the DNA phosphate backbone. Importantly, in the modified buffer, VirB no longer displayed any non-specific DNA-binding activity and only showed a marginal association with closed, *virS*-containing DNA when assayed in the absence of CTP. By contrast, strong binding was observed if CTP was included in the reaction (Fig. 4a). In line with the results of the nucleotide-binding assays (Fig. 2a–c), this interaction was strictly dependent on CTP and not observed in assays using CDP, UTP, ATP or GTP instead (Supplementary Fig. 6). Moreover, it required the presence of *virS*, because only residual binding was detected for similar DNA fragments lacking a *virS* sequence (Fig. 4b). A titration analysis showed that the CTP-dependent binding of VirB to *virS*-containing DNA occurred with high affinity (apparent $K_D = 2.3$ μM) (Fig. 4c, d). This behavior is highly reminiscent of the CTP-dependent loading of ParB onto *parS*-containing centromeric DNA, suggesting that VirB could use a similar loading-and-sliding mechanism to accumulate in promoter regions.

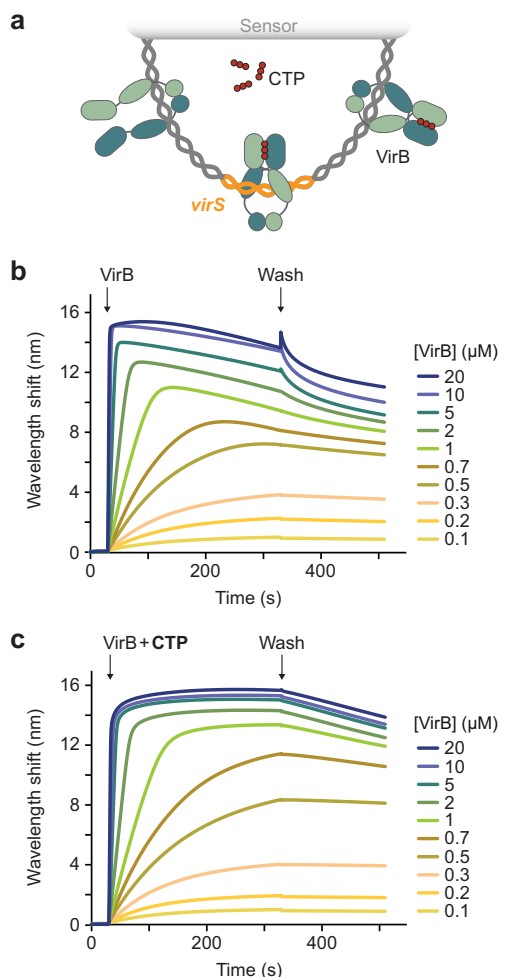

**Fig. 3 | CTP binding modulates the DNA-binding activity of VirB in low-stringency conditions. a** Schematic of the biolayer interferometry (BLI) setup used for the analyses in panels b and c. A double-biotinylated dsDNA fragment (215 bp) containing a central *virS* sequence (in orange) was immobilized on a streptavidin-coated biosensor and probed with VirB (green). **b, c** BLI analysis of the DNA-binding behavior of VirB in the **a** absence and **b** presence of CTP (1 mM) in low-stringency buffer (150 mM NaCl). Biosensors carrying the target DNA (at a density corresponding to a wavelength shift of ~1.3 nm) were probed with the indicated concentrations of VirB. At the end of the association reactions, the biosensors were transferred into protein- and nucleotide-free buffer to monitor the dissociation reactions (wash). The graphs show the results of a representative experiment ($n = 3$ independent replicates).

## VirB forms DNA-sliding clamps

Previous work has shown that ParB clamps cannot stably associate with *parS*-containing DNA molecules that have open ends, because they slide off the DNA as soon as they are released from their loading site[35,39]. To clarify the state of VirB after its CTP-dependent loading at *virS*, we therefore performed biolayer interferometry analyses of its interaction with open DNA fragments, again using stringent conditions that prevented non-specific DNA binding. For this purpose, we first probed biosensors carrying short (23 bp) double-stranded oligonucleotides with a central *virS* site that were only attached at one of their ends, so that the other end remained open (Fig. 5a). Subsequently, we analyzed the interaction of VirB with the immobilized DNA in the presence of CTP. Notably, even at very high concentrations, VirB barely associated with the open target DNA (Fig. 5b), even though the same oligonucleotides were densely covered with VirB in low-stringency conditions, which verifies the functionality of the biosensors used

(Supplementary Fig. 7). To further test for the ability of VirB to slide on DNA, we compared the interaction of VirB with biosensors carrying closed *virS*-containing DNA fragments (215 bp) before and after cleavage of the fragments with the restriction endonuclease NdeI (Fig. 5c). As expected, VirB strongly accumulated on closed target molecules in the presence of CTP. However, after opening of the fragments by NdeI treatment, the maximum binding levels were approximately fourfold lower, suggesting that VirB dimers are lost from the DNA after binding if its ends are no longer blocked (Fig. 5d). Moreover, during the association phase, the signal increased only briefly and then started to decrease, almost returning back to the baseline level. This behavior may be caused by the continuous loading and closure of VirB clamps at *virS*, which slide off the DNA and remain closed thereafter, leading to a steady decrease in the concentration of binding-competent, open VirB dimers.

The DNA-binding behavior of VirB suggested that it used a loading-and-sliding mechanism analogous to the one reported for ParB. We therefore aimed to determine whether the CTP-dependent interaction of VirB with *virS*-containing DNA could trigger the homo-dimerization of its two NBDs and thus close the VirB dimer into a ring-like structure, as also suggested by the structural model (Fig. 1b). For this purpose, we generated a fully functional (Supplementary Fig. 8) mutant variant of VirB (VirB-C5S/Q15C) that lacked the native cysteine residue at position 5 and carried an engineered cysteine residue at position 15, adjacent to the symmetry axis of the closed complex (Fig. 6a). Upon ring closure, the newly introduced C15 residues in the two NBDs were placed next to each other, enabling their covalent crosslinking by the bifunctional thiol-reactive compound bismaleimi-doethane (BMOE) (Fig. 6b). Using this approach, we observed that most VirB dimers remained in the open state when incubated alone or in the sole presence of *virS* DNA. However, upon the addition of CTP and, even more so, a combination of CTP and *virS* DNA, the proportion of closed complexes increased considerably (Fig. 6c, d). These findings support a model in which VirB forms DNA-sliding clamps that are loaded at *virS* sites and then closed by CTP-dependent homo-dimerization of the two NBDs.

Interestingly, similar results were obtained when the assay was performed with the wild-type protein, exploiting the native C5 residue for the crosslinking reaction (Supplementary Fig. 9). In this case, crosslinking was only mildly stimulated by CTP and largely dependent on the presence of both CTP and *virS*. According to the structural model, C5 is located in N-terminal helix of VirB, which is predicted to associate with the adjacent NBD. This arrangement would place the two C5 residues in the VirB dimer at a distance (39 Å) too large to allow their crosslinking by BMOE (8 Å length) (Supplementary Fig. 9a). The high efficiency of the crosslinking reaction thus suggests that the N-terminal helix may not be an integral part of the NBD but in a dynamic equilibrium between the bound and unbound state.

## CTP and *virS* DNA regulate the dynamics of VirB clamp closure

To further investigate the role of CTP and *virS* binding in the closure of VirB clamps, we analyzed the structural dynamics of VirB using hydrogen-deuterium exchange (HDX) mass spectrometry (Supplementary Data 1), a technique that detects local changes in the accessibility of backbone amide hydrogens caused by conformational changes or ligand binding[62]. The initial set of experiments was performed in low-stringency buffer (150 mM NaCl) (Supplementary Figs. 10a and 11a). Under these conditions, the addition of a double-stranded oligonucleotide containing a scrambled, non-functional *virS* sequence (Fig. 4b) led to a significant reduction in HDX in the C-terminal helices of the VBD as well as in the CTD compared to apo-VirB (Fig. 7a, c), consistent with the idea that these regions are responsible for the strong non-specific DNA-binding activity of VirB (compare Supplementary Fig. 2). The same regions exhibited reduced HDX in the presence of a *virS*-containing oligonucleotide, but in this

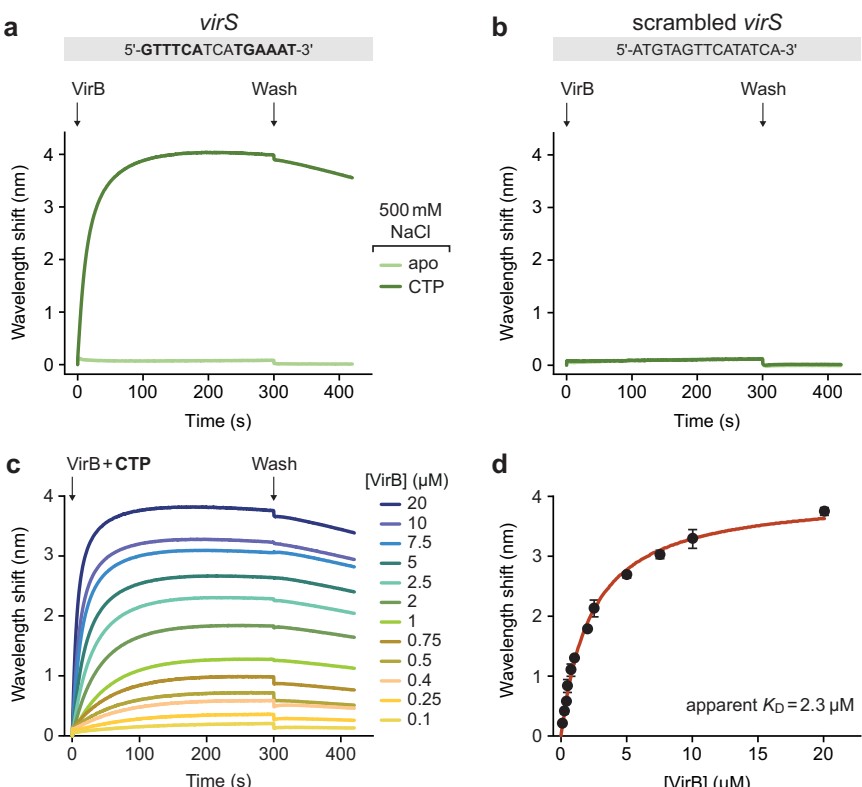

**Fig. 4 | VirB requires CTP- and *virS* binding to accumulate on DNA in high-stringency conditions. a** Biolayer interferometry analysis of the interaction of VirB with *virS*-containing DNA (215 bp) in high-stringency buffer (500 mM NaCl). Biosensors carrying a double-biotinylated, *virS*-containing DNA fragment (at a density corresponding to a wavelength shift of ~0.5 nm) were probed with VirB (20 μM) in the absence or presence of CTP (1 mM). The *virS* sequence used is shown at the top. The graph shows the results of a representative experiment (*n* = 3 independent replicates). **b** Same as in panel a, using a DNA fragment with a scrambled *virS* site. **c** Titration of double-biotinylated *virS*-containing DNA (215 bp) with increasing concentrations of VirB in the presence of CTP (1 mM) in high-stringency buffer (500 mM NaCl). DNA was immobilized as described in panel A. **d** DNA-binding affinity of VirB in high-stringency conditions. The maximal wavelength shifts measured at equilibrium in the traces shown in panel c were plotted against the corresponding VirB concentrations. Error bars indicate the SD (*n* = 3 independent replicates). A one-site-specific-binding model was used to fit the data. The calculated $K_D$ value is given in the graph. Source data are provided as a Source Data file.

case the changes in the VBD were more pronounced (Fig. 7a, c and Supplementary Figs. 11a and 12a). Moreover, *virS* DNA additionally induced a strong reduction in HDX in the N-terminal half of the VBD, harboring the HTH-motif responsible for *virS* recognition[25], as well as at the homodimerization interface of the NBD. Nucleotide-content analysis verified that the purified protein used for this analysis did not contain CTP or CDP (Supplementary Fig. 13). The juxtaposition of the two VBDs at the inverted repeats constituting the *virS* sequence thus appears to promote face-to-face interactions between the two NBDs independently of the presence of CTP. Notably, in the presence of *virS*, some peptides (e.g., residues 65–70 and 126–132) of the NBD showed a bimodal distribution of peptide ion intensities in their mass spectra, likely reflecting two populations with disparate HDX rates. This observation suggests that *virS*-bound VirB dimers dynamically switch between the open and closed state (Supplementary Fig. 14). In reactions containing only CTP, the differences in HDX observed in the NBD were even more pronounced than in the apo state and extended throughout the homodimerization interface and the nucleotide-binding pocket (Fig. 7a, c). Moreover, we observed reduced HDX in regions of the VBDs that are predicted to interact with each other in the closed complex, consistent with the idea that CTP binding promotes the closure of the VirB clamp and that this process is accompanied by rearrangements in VBDs that are likely to affect their DNA-binding behavior. However, again, peptides from the NBD showed a bimodal behavior, suggesting that the two NBDs are not stably associated with each other in the sole presence of CTP (Supplementary Fig. 14). Unlike in the case of CTP, the addition of CDP produced only minor shifts in

the HDX pattern of the nucleotide-binding pocket and the VBD (Supplementary Figs. 11a and 12b), as expected from its inability to promote the loading of VirB clamps onto DNA in vitro (Supplementary Fig. 6). Finally, in reactions containing both *virS* DNA and CTP, strongly reduced HDX was observed throughout all domains of VirB, including all regions that were affected in reactions containing only one of the two ligands (Fig. 7a, c), with a considerably higher amplitude of reduction in HDX than with CTP or *virS* alone. Moreover, in this case, all peptides that showed bimodal behavior with CTP as the only ligand were consistently shifted to the slow-exchanging state. In line with the crosslinking data (Fig. 6 and Supplementary Fig. 9), this observation indicates that a larger fraction of VirB dimers transitions to the ligand-bound closed state if both ligands are present. However, very similar results were also obtained for reactions containing both CTP and a double-stranded oligonucleotide with a scrambled *virS* sequence (Fig. 7c and Supplementary Fig. 12), suggesting that, in low-stringency conditions, non-specific DNA-binding may potentially be sufficient to stimulate VirB clamp closure.

Next, we performed the HDX analysis in high-stringency conditions (500 mM NaCl) that abolish non-specific DNA binding (Supplementary Figs. 10b and 11b). In this case, DNA did not produce a measurable change in the HDX pattern when assayed in the absence of CTP, even if it contained a *virS* site (Fig. 7b, c). This result suggests that, at elevated salt concentrations, *virS* binding is too dynamic to have a marked influence on the HDX reaction, consistent with the absence of a clear binding signal under similar conditions in the BLI assay (Fig. 4a). By, contrast, the addition of CTP again led to strong changes in the

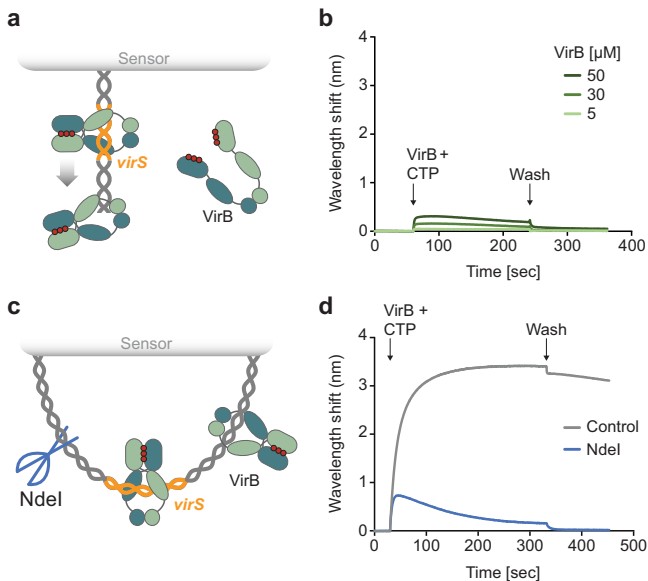

**Fig. 5 | VirB requires CTP- and *virS* binding to accumulate on DNA in high-stringency conditions. a** Biolayer interferometry (BLI) setup used to analyze the interaction of VirB with short, open *virS* DNA. A double-stranded *virS*-containing oligonucleotide biotinylated at one of its ends was immobilized on a streptavidin-coated biosensor. **b** BLI analysis using the setup described in panel a. The biosensors were probed with VirB at the indicated concentrations in the presence of CTP (1 mM), using high-stringency buffer (500 mM NaCl). **c** BLI setup used to compare the interaction of VirB with closed and open *virS* DNA. A double-biotinylated *virS*-containing DNA fragment (215 bp) was immobilized on two streptavidin-coated biosensors. Prior to the BLI assay, the biosensors either treated with the restriction endonuclease NdeI to open the immobilized DNA or incubated in the absence of NdeI as a control. **d** BLI analysis using the setup described in panel c. The biosensors incubated with or without NdeI were probed with VirB (20 μM) in the presence of CTP (1 mM), using high-stringency buffer (500 mM NaCl). The graphs in panels b and d show the results of representative experiments (*n* = 3 independent replicates each).

HDX pattern, similar to those observed in low-stringency conditions (Fig. 7b, c). Moreover, several peptides in the NBD (residues 63–68, 69–75, and 118–128) and the VBD (residues 218–238) again showed a bimodal behavior, suggesting that VirB alternated dynamically between the open and closed state in this condition (Supplementary Fig. 15). An additional, strong reduction in HDX was observed throughout all three domains of VirB when both CTP and *virS* were included in the reactions (Fig. 7b, c), consistent with the finding that both ligands are required to trigger robust VirB clamp closure in high-stringency buffer (Figs. 4a, b and 6 and Supplementary Fig. 9).

### VirB requires CTP binding to bind target DNA in vivo
Our analyses revealed that CTP was required to enable the specific loading of VirB clamps at *virS* sites in vitro. To clarify the role of CTP binding in the function of VirB in vivo, we made use of the R93A and R94A variants of VirB, which both lacked appreciable affinity for CTP (Fig. 2c). We confirmed that these variants were no longer loaded onto closed *virS*-containing DNA in high-stringency conditions (Fig. 8a) nor did they undergo CTP-dependent N-terminal homodimerization in crosslinking assays (Supplementary Fig. 16), indicative of a defect in CTP-mediated clamp closure. Inspired by work in *S. flexneri*[63], we then devised an in vivo assay that allowed us to visualize the recruitment of VirB to a *virS*-containing plasmid in the heterologous host *E. coli*, a species closely related to *S. flexneri*[64] that has been commonly used to investigate the mechanistic basis of gene regulation by VirB[47,49,50]. To this end, *E. coli* was transformed with a low-copy plasmid carrying the upstream region (187 bp) of the *S. flexneri icsB* gene, including the

previously reported *virS* site that mediates the regulation of the divergent *icsB* and *ipgD* promoters[47] (Supplementary Fig. 4). The resulting strain was then additionally transformed with expression plasmids that allowed the production of fluorescently (mVenus-) tagged versions of VirB or its two mutant variants under the control of an arabinose-inducible promoter (Fig. 8b and Supplementary Fig. 17a). Upon induction, -80% of the cells producing the wild-type mVenus-VirB fusion formed one to several bright foci per cell (Fig. 8c, d), reminiscent of results obtained previously for a synthetic *virS*-containing plasmid in pINV-free *S. flexneri* cells[63]. By contrast, only diffuse fluorescence was observed for cells producing the R93A or R94A variant, indicating that CTP binding is critical for the accumulation of VirB in the *icsB* promoter region in vivo (Fig. 8c, d). Notably, the wild-type protein failed to form foci in cells that contained a low-copy plasmid lacking the *icsB* upstream region instead of the original construct, consistent with the notion that the CTP-dependent loading of VirB in target promoter regions is strictly dependent on the presence of *virS* (Fig. 8c, d).

### The CTP-binding activity of VirB is critical for virulence gene expression in *S. flexneri*
Having identified a central role of CTP in the interaction of VirB with target promoter regions both in vitro and in a heterologous in vivo system, we went on to investigate the relevance of CTP binding to virulence gene expression in *S. flexneri* cells. First, we generated a mutant of the *S. flexneri* wild-type stain 2457T carrying a deletion of the *virB* gene. To this end, *virB* was replaced with an antibiotic resistance cassette that was flanked by FLP recombinase target (FRT) sites, using λ-Red recombineering[65]. Subsequently, FLP-mediated site-specific recombination was used to excise the cassette, leaving a single short FRT site in place of the *virB* coding sequence. The resulting strain (2457TΔ*virB*) was then used in complementation assays to investigate the effects of mutations in the CTP-binding site on the functionality of VirB in its native context. First, we performed localization studies to verify the deleterious effect of the R93A and R94A exchanges on the formation of VirB nucleoprotein complexes in *S. flexneri* cells (Fig. 9a, b). The corresponding mutant mVenus-VirB variants (Supplementary Fig. 17b) indeed showed a diffuse localization, whereas the wild-type fusion formed one or multiple distinct foci within the cell, confirming that VirB requires CTP binding to associate with target promoter regions in the endogenous pINV virulence plasmid. Next, we used whole-proteome analysis to obtain a global view of the regulatory activity of the different VirB variants. Cells of strain 2457TΔ*virB* expressing a plasmid-borne copy of wild-type *virB* produced the whole array of known virulence factors (Fig. 9c). These included the previously described members of the VirB regulon, such as the outer membrane protease IcsP[51] and the anti-activator of type III secretion OspD1[50], as well as components of the T3SS and various secreted effectors encoded in the two divergent operons in the Entry Region of pINV[26] (Supplementary Fig. 4). Under secretion-inducing conditions, these effectors were efficiently translocated to the culture medium, yielding a secretion profile very similar to that of the *S. flexneri* wild-type strain (Fig. 9d). Cells producing the R93A and R94A variants, by contrast, showed a strong reduction in the levels of virulence proteins (Fig. 9e, f). Consistent with this observation, they failed to secrete effectors to the medium and exhibited a secretion profile indistinguishable from that of the Δ*virB* mutant or *S. flexneri* strain 2457O, which carries a pINV derivative lacking the majority of virulence genes[66] (Fig. 9d). Together with our in vitro results, these finding demonstrate that VirB uses a CTP-dependent switch mechanism to activate virulence gene expression in *S. flexneri*.

## Discussion
The identification of the ParB/Srx domain as a CTP-binding module has led to fundamental new insights into the function of ParAB*S* DNA-partitioning systems. However, there are various ParB-like proteins

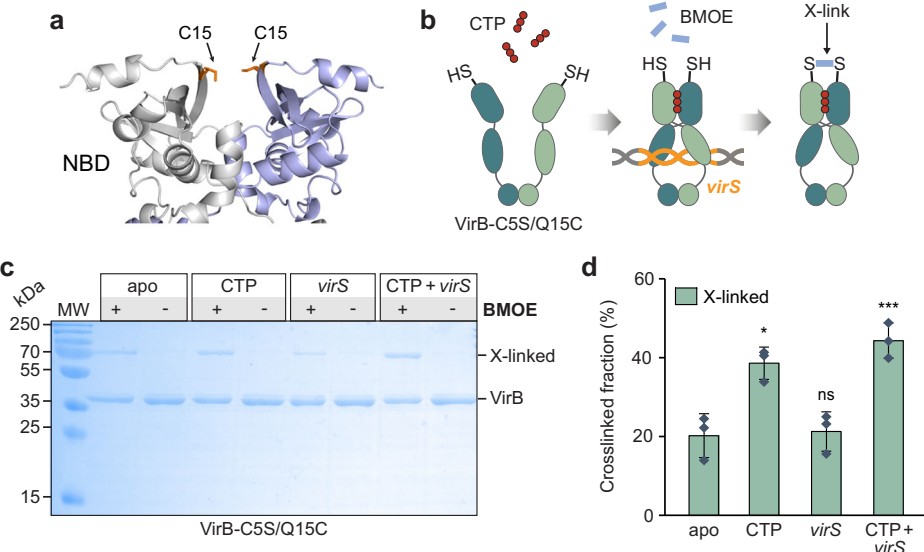

**Fig. 6 | VirB clamps close in the presence of CTP and *virS*-containing DNA in vitro. a** Model of the VirB-C5S/Q15C dimer based on an Alphafold-Multimer[55] prediction. The newly introduced C15 residues are shown in stick representation (predicted distance: 6.1 Å). **b** Schematic showing the crosslinking assay used to detect VirB clamp closure. The engineered C15 residue of VirB-C5S/Q15C is shown as a thiol group (-SH). The closure of the VirB clamp reduces the distance between the two C15 residues, thereby enabling their covalent crosslinking by the bifunctional thiol-reactive crosslinking agent BMOE. **c** SDS-gel showing the protein species obtained in the in vitro crosslinking analysis. VirB-C5S/Q15C was incubated for 5 min alone, with CTP (1 mM), with a double-stranded DNA oligonucleotide containing a *virS* motif (1 μM; virS-icsB-for/virS-icsB-rev) or with both CTP and *virS* DNA prior to crosslinking with BMOE and analysis of the reaction products by SDS-PAGE. Monomeric VirB and the dimeric crosslinking product (X-linked) are indicated. MW: Molecular weight marker. **d** Quantification of the fractions of crosslinked protein obtained in the indicated conditions. The columns display the mean (±SD) of three independent measurements (diamonds). *$p = 0.012$, ***$p = 0.0049$, ns not significant ($p = 0.82$) (unpaired two-sided Welch's *t*-test; compared to the apo state). Source data are provided as a Source Data file.

that are not encoded in *parABS* operons and thus likely to mediate processes other than DNA segregation. Only very few representatives of these orphan ParB homologs have been investigated to date. One of them is the nucleoid occlusion protein Noc of *Bacillus subtilis*, a close homolog of chromosomally encoded ParB proteins that has recently been shown to use a CTP-dependent clamping-and-sliding mechanism to accumulate on chromosomal DNA and tether it to the cytoplasmic membrane[67], thereby preventing the assembly of the cell division apparatus over the nucleoid[68,69]. Another prominent member of this group is VirB, which has evolved into a transcriptional regulator with a critical role in *S. flexneri* virulence gene expression.

Structural modeling suggests that VirB is derived from plasmid-encoded ParB proteins and forms clamp-like dimeric structures that are closed by homodimerization of the NBDs and CTDs (Fig. 1). In support of this model, previous work has shown that the two CTDs closely associate with each other and stably connect the two VirB subunits at their C-terminal ends[25]. Moreover, truncations of the NBD or CTD were found to completely abolish the function of VirB, underscoring the relevance of clamp formation and closure for its regulatory activity[24]. The NBD of VirB is highly conserved and binds CTP with similar or even higher affinity than previously characterized ParB homologs[38,39], with a clear preference for CTP over CDP (Fig. 2a, b). Given the relatively high concentration of CTP in the cytoplasm (~500 μM)[56], the nucleotide-binding site of VirB is thus likely to be saturated with CTP at all times. Our biolayer interferometry, crosslinking and HDX analyses clearly demonstrate that CTP-binding facilitates the transition of VirB clamps from an open to a closed state in which they embrace target DNA in a ring-like fashion. As in the case of ParB, clamp closure by homodimerization of the CTP-bound NBDs is predicted to entail a crossover of the two polypeptide chains, positioning the NBD of one subunit next to the VBD of the respective *trans*-subunit (Fig. 1b). Importantly, we observe that CTP-mediated VirB clamp closure is stimulated by *virS* DNA (Figs. 6 and 7). In low-stringency conditions, the addition of *virS* alone was sufficient to induce changes in the HDX pattern of the NBD

that are indicative of transient NBD homodimerization events. This observation suggests that the juxtaposition of the VBDs at the *virS* site facilitates the face-to-face interaction of the two NBDs, likely by increasing their spatial proximity. The sole presence of CTP, by contrast, led to marked global changes in HDX that point to a more stable, but still transient association of the NBDs. Only if both *virS* and CTP were provided, VirB clamps closed robustly and with maximal efficiency. Notably, the crosslinking and HDX behavior of VirB in the presence of CTP is reminiscent of the behavior of ParB in the presence of CTPγS[37,39], consistent with the observation that VirB shows exceedingly weak CTPase activity under all conditions tested.

The strong non-specific DNA-binding activity of VirB complicates the analysis of its interaction with target promoters in vitro. In low-stringency buffer, VirB accumulated on DNA independently of the presence of CTP and *virS*, although DNA binding was more robust in the presence of CTP (Fig. 3). Moreover, when analyzed in reactions containing CTP, non-specific DNA and *virS* DNA produced essentially the same changes in the HDX pattern of VirB (Fig. 7a and Supplementary Fig. 12a). This observation suggests that low-stringency conditions may allow *virS*-independent clamp closure, although it remains to be clarified whether this process occurs efficiently at cytoplasmic VirB and ligand concentrations. By contrast, in more stringent conditions that reduce the effect of non-specific protein-DNA interactions, VirB clamps are loaded specifically at *virS* sites, in a process strictly dependent on the presence of CTP (Fig. 4). As observed for ParB, closed clamps are released from *virS* and slide laterally along the DNA, enabling the loading of multiple VirB dimers at a single *virS* site (Fig. 5). Consistent with results previously obtained in *S. flexneri*[63], our in vivo analysis confirmed that the presence of a single *virS* site is indeed sufficient to recruit a large fraction of VirB molecules to a *virS*-containing plasmid in *E. coli* cells (Fig. 8). This process was abolished by the mutation of residues essential for CTP binding, verifying the critical relevance of CTP-dependent clamp closure for the association of VirB with target promoter regions. It still remains to be clarified how VirB can efficiently interact with *virS* sites

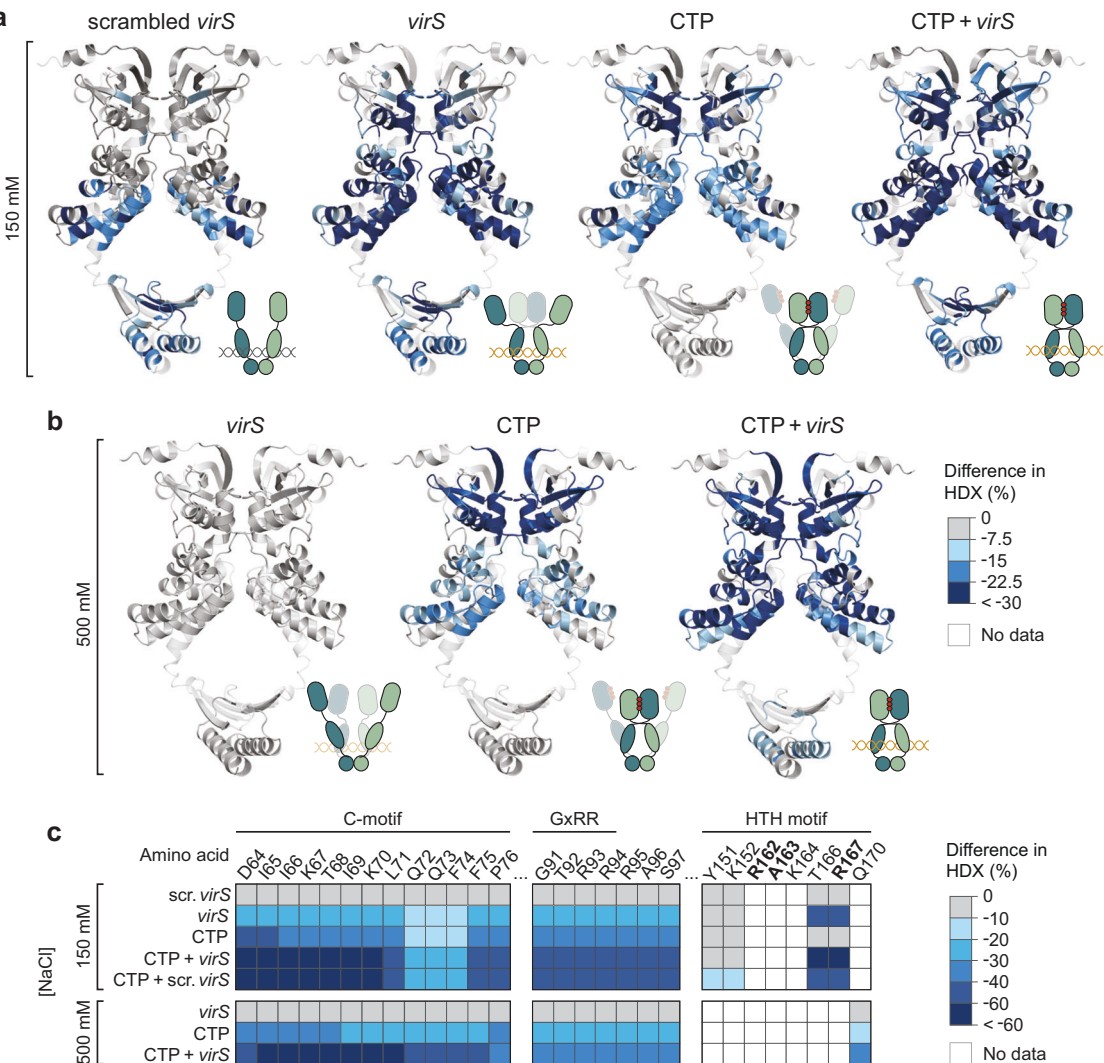

**Fig. 7 | CTP and *virS* DNA cooperatively stimulate the homodimerization of the N-terminal region of VirB. a**, **b** Hydrogen-deuterium exchange (HDX) mass spectrometry analysis of VirB in the presence of different ligands. VirB was incubated with an equimolar concentration of double-stranded DNA oligonucleotides containing a scrambled (scrambled-virS-for/scrambled-virS-rev) or intact (virS-icsB-for/virS-icsB-rev) *virS* motif and/or CTP (10 mM) in **a** low-stringency or **b** high-stringency buffer. Shown are the maximal differences in HDX obtained in the indicated conditions compared to the apo state, projected onto the AlphaFold-Multimer model of the VirB dimer. The color code is given in panel b. Blue color indicates regions that show reduced HDX upon ligand binding. The schematics next to the structural models indicate the most likely conformational state of the VirB dimer in the respective conditions. Protein regions not covered by any peptides are displayed in transparent white. **c** Heatmap of the maximal differences in HDX obtained in the indicated conditions for representative residues in the conserved C-, GxRR, and HTH motifs of VirB. The color code is given on the right. A detailed report of the HDX analysis is given in Supplementary Data 1.

despite the large excess of non-specific DNA within the cell. On the one hand, the lower salt concentration in the cytoplasm may be compensated by its high content of organic compounds and macromolecules, which could potentially also block the positively charged regions of VirB and, thus, reduce its non-specific DNA-binding activity. On the other hand, the affinity of VirB for *virS* sites is likely to be markedly higher than that for non-specific DNA, since *virS* binding involves both non-specific interactions with the DNA backbone and specific interactions with the HTH-motif[25]. In support of this notion, previous work has shown that even a large (>50,000-fold) excess of non-specific DNA is not sufficient to prevent the formation of specific VirB-*virS* complexes[25]. Another open question concerns the mechanism that controls the dynamics of VirB clamp opening in the absence of appreciable CTPase activity. In the case of ParB, nucleotide hydrolysis was shown to serve two important purposes. On the one hand, it re-opens prematurely closed clamps to ensure the quantitative loading of ParB dimers onto DNA. On the other hand, it triggers robust clamp opening after loading and thus promotes the dissociation of VirB dimers from their target DNA, thereby

determining the sliding time of ParB clamps and their degree of spreading within the centromere region[34,36,37]. However, at least in *M. xanthus*, CTPase-deficient ParB variants are still quantitatively loaded onto DNA, and they are still confined to a defined region within the centromere, although their longer sliding times lead to considerable increase in their spreading distances[37]. This observation is explained by spontaneous, CTPase-independent dissociation of the NBDs, which enables the release of ParB clamps without nucleotide hydrolysis, albeit at relatively low rates[34,37]. Given its weak CTPase activity, VirB may rely on a similar mechanism to control its distribution within target promoter regions, and it will be interesting to study its spreading behavior and the kinetics of its release from DNA in vivo.

How can the loading and spreading of VirB clamps at *virS* sites affect promoter activity? VirB stimulates the expression of target genes by counteracting their silencing by H-NS[70], a nucleoid-associated protein that can bridge DNA and thus stabilize negatively supercoiled DNA regions[71,72]. Interestingly, the interaction of VirB with target promoter regions has recently been shown to generate torsional stress in

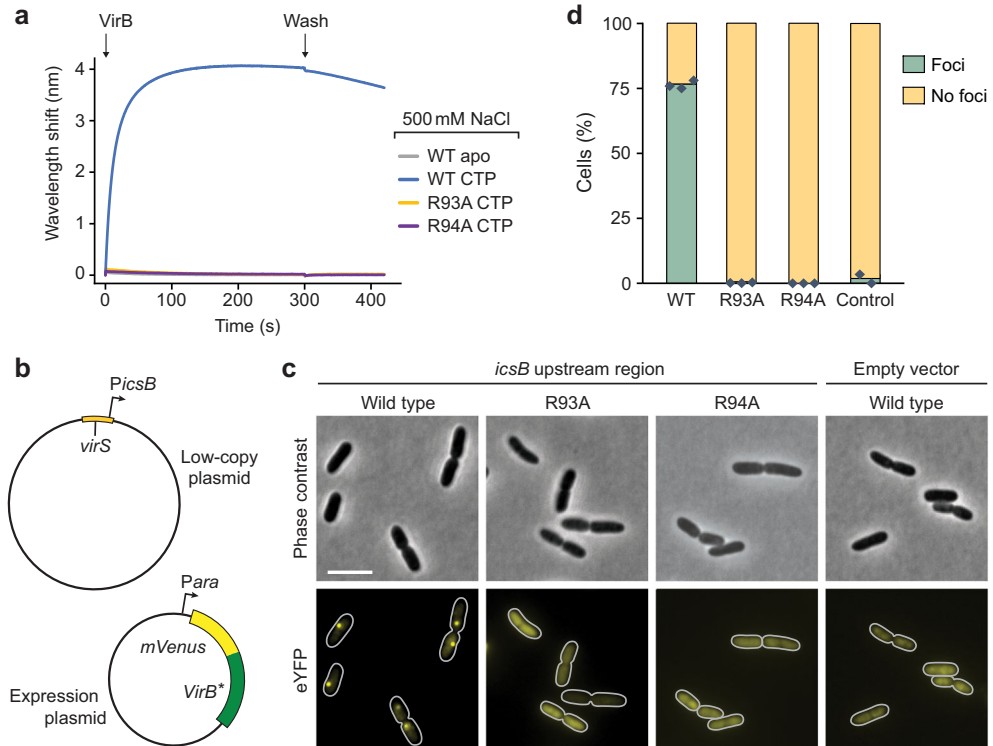

**Fig. 8 | CTP binding is critical for the loading of VirB on *virS*-containing DNA in vivo. a** Biolayer interferometry analysis of the interaction of VirB-R93A and VirB-R94A with a closed, *virS*-containing DNA fragment (215 bp) in high-stringency buffer (500 mM NaCl). After derivatization with the double-biotinylated *virS* DNA, the biosensor was probed with wild-type VirB or its mutant derivatives (20 μM) in the absence (apo) or presence of CTP (1 mM). **b** Plasmids used for the in vivo binding assay. **c** Localization patterns of wild-type or mutant mVenus-VirB fusions in *E. coli* TOP10 derivatives that harbor low-copy plasmids with or without the *icsB* upstream region. Cells carrying a low-copy containing (pSJ30) or lacking (pSJ31) the

*icsB* upstream region were transformed with expression plasmids that allow the production of the indicated mVenus-VirB variants under the control of an arabinose-inducible promoter (pSJ18, pSJ20, pSJ21). Transformants were induced with 0.1% (w/v) arabinose for 4 h prior to analysis by fluorescence microscopy. Scale bar: 4 μm. **d** Quantification of the proportion of cells showing distinct foci in the experiment described in panel b. Bars indicate the mean of 2–3 biological replicates (diamonds). Number of cells analyzed in total: WT (1857), R93A (1818), R94A (1479), WT without *virS* (1477).

target DNA molecules that decreases their degree of negative supercoiling in vivo[53], thereby potentially destabilizing the H-NS nucleoprotein complexes that block transcription initiation at VirB-dependent promoters (Fig. 10). This process may be facilitated by topoisomerase I, which helps to relax excess negative supercoiling in promoter regions during gene transcription[73,74]. Consistent with this idea, the anti-silencing activity of VirB was found to be attenuated in cells lacking this enzyme[53,75]. The contribution of topoisomerase I may be particularly important if single *virS* sites mediate the regulation of two flanking, divergent transcription units, as shown for example for the two large operons in the Entry Region of pINV (Supplementary Fig. 4), because this arrangement hinders the diffusion of negative supercoils away from the adjacent promoters. In addition to causing torsional stress, the spreading of VirB clamps within promoter regions could sterically hinder the alignment and bridging of DNA by H-NS, thereby reinforcing its direct effect on DNA topology. Given that the spreading of both ParB[33,35,39] and VirB[49] clamps can be hindered by strong roadblocks, it is tempting to speculate that the anti-silencing activity of VirB could be modulated by additional regulatory proteins that bind *virS*-containing promoter regions to fine-tune the timing of VirB-dependent gene expression.

The precise mechanism by which VirB clamps modulate DNA supercoiling remains to be determined. It is conceivable that the two VBDs of a VirB dimers are bound slightly out of phase, so that their alignment upon clamp closure induces a small rotation of the associated *virS* half-sites in opposite directions, underwinding the DNA that is trapped within the clamp and slightly overwinding the flanking DNA regions. The strong non-specific DNA-binding activity may enable VirB

to remain in close contact with the DNA after leaving *virS* and thus maintain this torsional force during the sliding process. The loading of many VirB clamps and their spreading within the promoter region may amplify the torsion generated and expand the size of the overwound region, thereby leading to a local relaxation of negative supercoils. Notably, recent in vitro and modeling studies have suggested that sliding ParB clamps can interact face to face to bridge spatially adjacent DNA regions[46,76,77]. It is conceivable that VirB clamps also exhibit such a bridging activity, which could contribute to modulate the topology of *virS*-containing promoter regions. However, detailed structural studies are required to test this hypothesis.

Collectively, our work shows that VirB forms a distinct group of ParB-like proteins that uses a CTP-dependent switch mechanism to associate with target promoter regions and activate gene expression. Future work will be required to fully unravel the connection between the loading and sliding of VirB clamps and their antagonistic effect on H-NS-mediated gene silencing. Given the diverse biological activities of orphan ParB homologs, it will be interesting to study more members of this intriguing group of proteins and determine the full breadth of functions they fulfill. Moreover, it is tempting to speculate that the distinctive nucleotide-binding domain and switch mechanism of VirB could be exploited for the development of antibacterial drugs that specifically suppress the induction of the *S. flexneri* virulence program.

## Methods
### Plasmids, strains, and growth conditions
The plasmids, strains, and oligonucleotides used in this study are listed in Supplementary Tables 1–3. The sequences of all plasmids

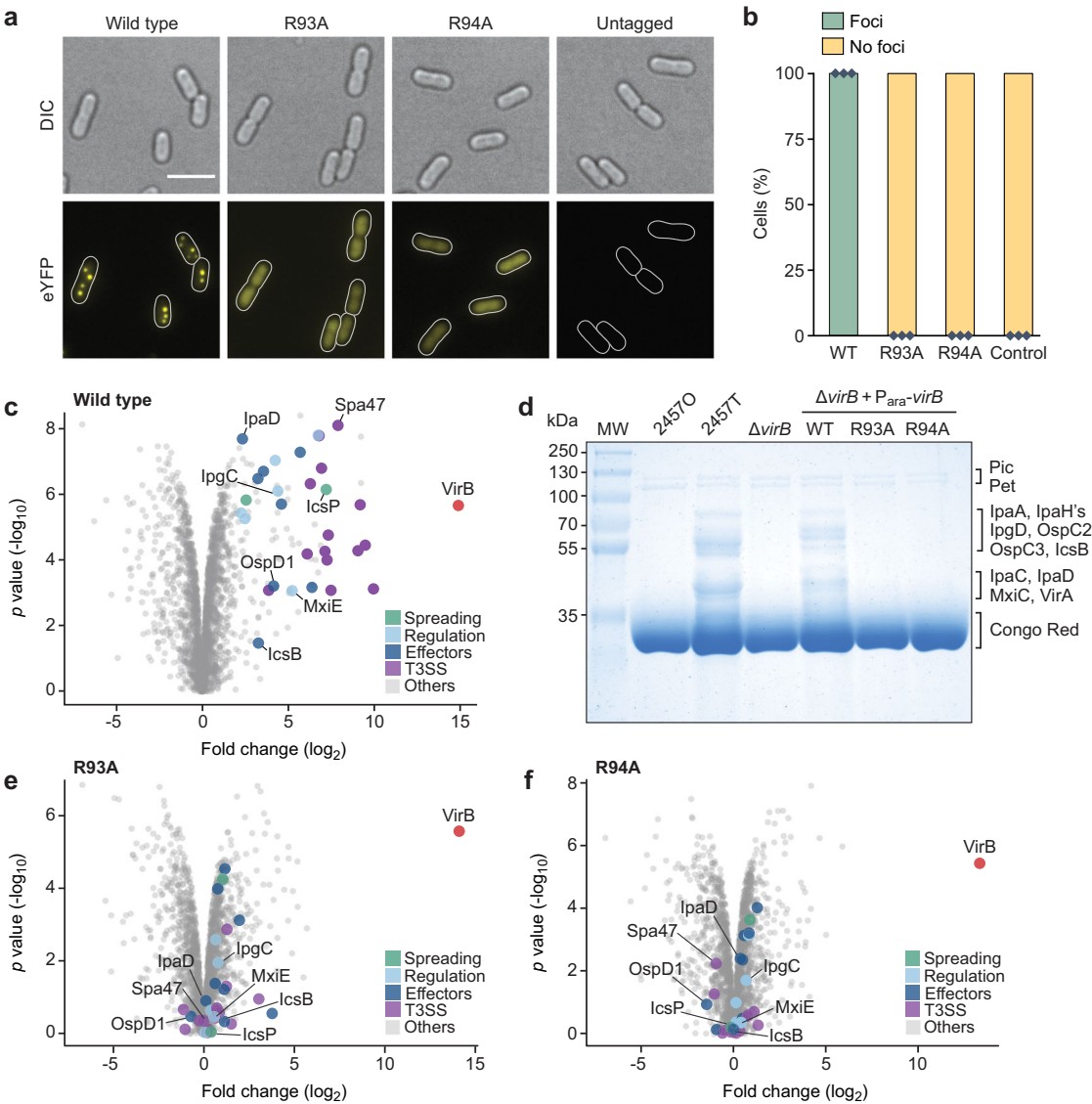

**Fig. 9 | Virulence gene expression in *S. flexneri* depends on the CTP-binding activity of VirB. a** Localization of wild-type or mutant mVenus-VirB fusions in *S. flexneri* under virulence-inducing conditions. *S. flexneri* 2457TΔ*virB* was transformed with plasmids encoding the indicated mVenus-VirB fusions (pSJ37, pSJ38, pSJ39) or wild-type VirB (pSJ27) (untagged) under the control of an arabinose-inducible promoter. The transformants were shifted to 37 °C and grown for 2 h in the presence of arabinose prior to imaging. Scale bar: 4 μm. **b** Quantification of the proportion of cells showing distinct foci in the experiment described in panel a. Bars indicate the mean of 3 biological replicates (diamonds). Number of cells analyzed in total: WT (1718), R93A (1577), R94A (1625), untagged VirB (1681). **c** Total proteome analysis showing the induction of virulence gene expression in *S. flexneri* cells producing wild-type VirB. *S. flexneri* strain 2457TΔ*virB* was transformed with a plasmid that carries wild-type *virB* under the control of an arabinose-inducible promoter (pJS27). The resulting transformant and the plasmid-free parental strain were shifted to 37 °C, cultivated in the presence of arabinose and Congo Red and subjected to total proteome analysis. Each data point in the Volcano plot represents a different protein. The x-coordinate indicates the $\log_2$ of the average difference in the peptide counts for a given protein between the VirB-producing strain and the plasmid-free parental reference strain. The y-coordinate gives the $-\log_{10}$ of the corresponding *p* value (unpaired two-sided *t*-test). Known virulence factors are highlighted in color. Representative proteins are labeled. Data represent the average of three independent biological replicates. A detailed list of the data obtained in the proteomics analyses is provided in Supplementary Data 2. **d** Secretion profiles of various *S. flexneri* strains. Strains 2457O (avirulent), 2457T (wild type), 2457TΔ*virB* and derivatives of 2457TΔ*virB* bearing plasmids that carry the indicated *virB* alleles under the control of an arabinose-inducible promoter (pSJ27, pSJ28, pSJ29) were shifted to 37 °C and cultivated in the presence of arabinose and Congo Red. Subsequently, secreted proteins were precipitated from the culture supernatant and analyzed by SDS-PAGE. Note that Congo Red stimulates the secretion of proteins through the T3SS by mimicking the natural trigger provided by the interaction of cells with the host cell membrane[88]. The gel shows the result of a representative experiment (*n* = 3 independent replicates). Details on the mass spectrometry-based assignment of the protein bands are given in Supplementary Data 3. Total proteome analyses of *S. flexneri* 2457O and 2457T are shown in Supplementary Fig. 18. **e**, **f** Total proteome analyses showing the lack of virulence gene induction in *S. flexneri* cells producing the CTP-binding-defective **e** R93A and **f** R94A variants of VirB. The experiments were performed as described in panel c, using *S. flexneri* Δ*virB* cells transformed with plasmids that carry the indicated mutant *virB* alleles (pJS28, pSJ29).

were verified by DNA sequencing using SnapGene 3.3.4 software (GLS Biotech, USA). *E. coli* was cultivated at 37 °C in Luria-Bertani (LB) medium, supplemented with ampicillin (200 μg/mL) and/or chloramphenicol (34 μg/mL) when appropriate. *S. flexneri* was grown

in LB medium, supplemented with ampicillin (100 μg/mL) when appropriate.

A mutant of *S. flexneri* 2457T carrying an in-frame deletion in *virB* (2457TΔ*virB*) was generated using a modified λ-Red-based

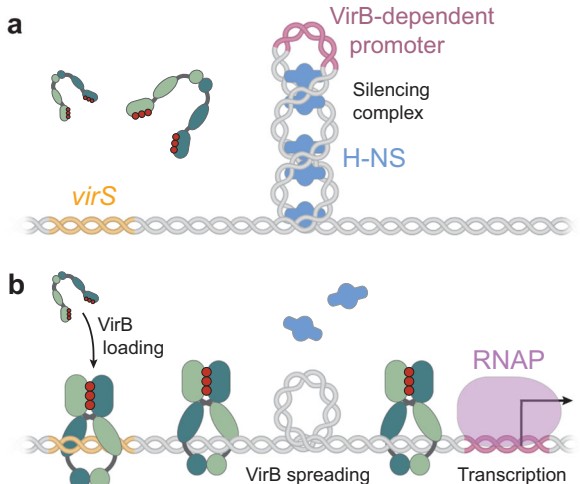

**Fig. 10 | Hypothetical model of the mechanism underlying VirB-dependent gene regulation. a** Before VirB associates with the virulence plasmid, the nucleoid-organizing protein H-NS binds and thus stabilizes negative DNA supercoils in the promoter regions of VirB-regulated genes, thereby sequestering the promoters from RNA polymerase and silencing gene expression. **b** The CTP-dependent loading of VirB clamps at *virS* sites and their spreading into the adjacent promoter regions leads to local overwinding of the DNA. This effect may destabilize the adjacent H-NS nucleoprotein complexes and reduce the degree of negative supercoiling in the vicinity of the promoter, thereby making it accessible to RNA polymerase and allowing transcription to occur.

recombineering approach[65]. Briefly, a Kan$^R$ resistance cassette flanked by FRT recombination sites was PCR-amplified from plasmid pCLF4[78] with primers AD1914 and AD1915, which included the upstream and downstream flanking regions of *virB*, respectively. After purification, the reaction product (1 μg) was transferred into *S. flexneri* 2457T cells bearing plasmid pKD46[65] by electroporation. Kanamycin-resistant clones carrying the resistance cassette in place of the *virB* gene were verified by colony PCR and then transformed with plasmid pCP20[79] to induce the removal of the resistance cassette by FLP-mediated site-specific recombination at the flanking FRT sites. The resulting markerless mutant cells were then cured of the temperature-sensitive plasmids pCP20 and pKD46 by overnight cultivation at 41 °C. The absence of the plasmids was confirmed by testing for sensitivity against the respective antibiotics.

## Protein overproduction and purification

Wild-type VirB or mutant derivatives carrying an N-terminal His$_6$-SUMO tag[80] were overproduced in *E. coli* Rosetta(DE3)pLysS transformed with pSJ01, pSJ02, pSJ05, pSJ013, or pSJ014. Cultures were grown at 37 °C in 3 L LB medium with ampicillin and chloramphenicol to an $OD_{600}$ of 0.6, induced to overproduce the protein of interest by the addition of 1 mM isopropyl-β-thiogalactopyranoside (IPTG), and further incubated at 18 °C overnight. The cells were harvested by centrifugation at $10,000 \times g$ for 20 min at 4 °C and resuspended in lysis buffer (25 mM HEPES/NaOH pH 7.5, 500 mM NaCl, 0.1 mM EDTA, 5 mM $MgCl_2$). After another centrifugation step at $6500 \times g$ for 30 min at 4 °C, the washed cells were resuspended in buffer A (25 mM HEPES/NaOH pH 7.5, 500 mM NaCl, 0.1 mM EDTA, 5 mM $MgCl_2$, 30 mM imidazole, 1 mM β-mercaptoethanol) supplemented with 10 mg/mL DNase I and 100 μg/mL phenylmethylsulfonyl fluoride and disrupted by three passages through a French press (16,000 psi). The cell lysate was centrifuged at $30,000 \times g$ for 30 min at 4 °C, and the supernatant was filtered through a syringe filter with a pore size of 0.2 μm. Subsequently, proteins were separated by immobilized-metal affinity chromatography (IMAC) on a 5 mL HisTrap HP column (GE Healthcare), previously washed with ddH$_2$O and equilibrated with buffer B

(25 mM HEPES/NaOH pH 7.5, 1 M NaCl, 0.1 mM EDTA, 5 mM $MgCl_2$, 30 mM imidazole, 1 mM β-mercaptoethanol). Protein was eluted at a flow rate of 1 mL/min with a linear gradient from 30 mM to 300 mM imidazole, obtained by mixing buffer B with buffer C (25 mM HEPES/NaOH pH 7.5, 500 mM NaCl, 0.1 mM EDTA, 5 mM $MgCl_2$, 300 mM imidazole, 1 mM β-mercaptoethanol). Eluate fractions were analyzed by SDS-PAGE, and fractions containing the protein of interest in high concentration and purity were pooled and dialyzed overnight against 3 L of buffer D (25 mM HEPES/NaOH pH 7.5, 500 mM NaCl, 0.1 mM EDTA, 5 mM $MgCl_2$, 10% (v/v) glycerol, 1 mM β-mercaptoethanol). After the removal of precipitates by centrifugation at $30,000 \times g$ for 30 min at 4 °C, the protein solution was filtered as described above. Subsequently, the His$_6$-SUMO tag was removed by treatment with Ulp1 protease[80] in the presence of 1 mM DTT, while the protein solution was dialyzed overnight against buffer D. A second IMAC step was then performed to separate the cleaved His$_6$-SUMO tag from the protein of interest. Suitable flow-through fractions were pooled and dialyzed overnight against buffer D. After concentration to a final volume of 5 mL, the solution was subjected to size-exclusion chromatography (SEC) on a HighLoad 75 Superdex column (GE Healthcare) equilibrated with buffer D. Fractions containing pure protein of interest were pooled, concentrated, snap-frozen in liquid N$_2$ and stored at −80 °C until further use.

*M. xanthus* ParB was purified as described previously[37].

## Isothermal titration calorimetry

Nucleotide-binding assays using isothermal titration calorimetry were performed with a MicroCal PEAQ-ITC system (Malvern Panalytical, USA) at 25 °C. Prior to the measurements, CTPγS (custom synthesized by Jena Biosciences, Germany) and CDP were dissolved to a concentration of 1.55 mM in reaction buffer (25 mM HEPES/NaOH pH 8, 150 mM NaCl, 0.1 mM EDTA, 5 mM $MgCl_2$). Subsequently, the nucleotide solutions were titrated to 115 μM VirB in 13 consecutive injections (2 μL), performed at 150-s intervals with a duration of 4 s per injection. The mean enthalpies of dilution were subtracted from the raw titration data before analysis. The titration curves obtained were fitted to a one-set-of-sites model using the MicroCal PAEQ-ITC analysis software (Malvern Panalytical).

## Microscale thermophoresis

Nucleotide-binding assays based on microscale thermophoresis were performed with a Monolith NT.115 instrument (NanoTemper Technologies GmbH, Germany) using Monolith NT Premium Capillaries. Proteins were fluorescently labeled using the RED-Maleimide 2nd Generation Protein Labeling Kit (NanoTemper Technologies GmbH, Germany) as recommended by the manufacturer. 50 nM labeled protein was then mixed with CTP, CDP or ATP at concentrations ranging from 61 nM to 2 mM in MST buffer (25 mM HEPES/NaOH pH 7.2, 500 mM NaCl, 5 mM $MgCl_2$, 7.5% (v/v) glycerol, 0.06% [v/v] Tween 20). Measurements were performed at 25 °C, with the red LED laser adjusted to a power of 50% (VirB-R93A) or 70% (VirB and VirB-R94A) and the infrared laser set to 50%. Two to three independent measurement (three technical replicates each) were performed for each condition. Data were analyzed using MO Affinity Analysis v2.3 (NanoTemper Technologies GmbH, Germany). To prevent heat-induced artifacts and, at the same time, avoid analyzing only the signal change induced by the initial temperature jump, the following regions were used for data analysis: cold region from -1 s to 0 s, hot region from 1.5 s to 2.5 s.

## CTPase assay

Nucleotide hydrolysis was analyzed using an NADH-coupled enzyme assay[81,82]. The reactions included 5 μM protein (VirB or ParB), 1 mM CTP, 800 μg/mL NADH, 20 U/mL L-lactate dehydrogenase (Sigma−Aldrich), 20 U/mL pyruvate kinase (Sigma−Aldrich) and 3 mM PEP in a buffer

composed of 25 mM HEPES/NaOH pH 7.2, 5 mM MgCl₂, 150 mM NaCl and 1 mM DTT. When required, 0.5 µM of a double-stranded DNA oligonucleotide (23 bp) containing a *virS* site (virS-icsB-for/virS-icsB-rev) or a scrambled *virS* site (Scrambled-virS-for/Scrambled-virS-rev) or 0.3 µM of a DNA stem-loop (54 bases; *parS*-Mxan-wt) containing a wild-type *parS* site were added to the reactions (Table S3). After the start of the reactions by the addition of CTP, 150 µL of each mixture were transferred to a 96-well plate (Sarstedt, Germany). Subsequently, the absorbance of NADH at a wavelength of 340 nm was measured over 60 min at 2-min intervals in an Epoch 2 microplate spectrometer (Bio-Tek Instruments, Germany), which was set to a temperature of 30 °C. A control reaction lacking CTP was analyzed to correct for NADH oxidation and background activity. Data were recorded with Gen5 2.07.17 (BioTek Instruments, Germany) and analyzed using Excel 2019 (Microsoft). The data points were fitted to a linear equation, whose slope was then used to calculate the turnover numbers of VirB and ParB for CTP.

### Biolayer interferometry

Biolayer interferometry (BLI) analyses were performed using a Bli(tz) system (ForteBio, Pall Life Science), equipped with High Precision Streptavidin (Octet SAX2) biosensors (Sartorius, USA) that were derivatized with biotinylated DNA fragments, Short, single-biotinylated DNA fragments (23 bp) were assembled from two oligonucleotides (Eurofins, Germany), which were mixed, heated to 95 °C for 5 min and then annealed by gradual reduction of the temperature. Long, double-biotinylated DNA fragments (215 bp) were obtained by PCR amplification of custom-synthesized DNA fragments (Strings™ DNA Fragments; Invitrogen, Germany) with biotinylated primers (Bio-icsB-for/Bio-icsB-rev), followed by gel purification. Reactions were carried out in low-stringency (25 mM HEPES/NaOH pH 7.5, 150 mM NaCl, 5 mM MgCl₂, 1 mM DTT, 10 µM BSA, 0.01% [v/v] Tween 20) or high-stringency (25 mM HEPES/NaOH pH 7.5, 500 mM NaCl, 5 mM MgCl₂, 1 mM DTT, 10 µM BSA, 0.01% [v/v] Tween 20) binding buffer. After the establishment of a stable baseline, the association reactions were monitored at different concentrations of VirB in binding buffer. To monitor the dissociation kinetics, the sensor was subsequently transferred to a protein-free buffer. Data were recorded with BLItz Pro 1.2.1.5 (ForteBio, Pall Life Science) and analyzed using Excel 2019 (Microsoft).

To analyze VirB for sliding behavior, the closed 215-bp *virS*-containing DNA fragment was opened by cleavage with NdeI (NEB, Germany). To this end, biosensors with immobilized DNA were incubated overnight at 37 °C with 80 U NdeI in 300 µL rCutSmart buffer (NEB, Germany). In parallel, a control biosensor was prepared by incubation in 300 µL rCutSmart buffer in the absence of NdeI. Subsequently, the biosensors were probed with VirB in high-stringency buffer as described above.

### Hydrogen-deuterium exchange (HDX) mass spectrometry

HDX-MS experiments were carried out similarly as described previously[37,38] with minor modifications. In HDX experiment 1 (low-stringency buffer), the samples contained 25 µM VirB in a buffer composed of 25 mM HEPES-NaOH pH 7.5, 150 mM NaCl, 5 mM MgCl₂, 0.1 mM EDTA and 1 mM DTT. In HDX experiment 2 (high-stringency buffer), the samples contained 50 µM VirB in a buffer composed of 25 mM HEPES-NaOH pH 7.5, 500 mM NaCl, 5 mM MgCl₂, 1 mM DTT and 0.1 mM EDTA. When indicated, double-stranded DNA oligonucleotides containing a native or scrambled *virS* sequence were used at the same concentration as VirB, and CTP was used at a final concentration of 10 mM.

HDX reactions were prepared automatically with a two-arm robotic autosampler (LEAP Technologies). 7.5 µL of protein solution was dispensed in a glass well plate kept at 25 °C and supplemented with 67.5 µL of buffer (see above) prepared in 99.9% D₂O to initiate the hydrogen/deuterium exchange reaction. After incubation for 10, 30,

100, 1000, or 10,000 s, 55 µL of the reaction were withdrawn and added to 55 µL of quench buffer (400 mM KH₂PO₄/H₃PO₄, 2 M guanidine-HCl, pH 2.2), which was pre-dispensed in another glass well plate cooled at 1 °C. Following mixing, 95 µL of the resulting mixture were injected into an ACQUITY UPLC M-Class System with HDX Technology through a 50 µL sample loop. Non-deuterated samples were prepared using a similar procedure by 10-fold dilution in buffer prepared with H₂O followed by an ~10 s incubation at 25 °C prior to quenching and injection. The samples were flushed out of the sample loop with water +0.1% (v/v) formic acid (100 µL/min flow rate) over 3 min and guided to a cartridge (2 mm × 2 cm) that contained immobilized porcine pepsin for proteolytic digestion at 12 °C. The resulting peptic peptides were collected on a trap column cartridge (2 mm × 2 cm) that was filled with POROS 20 R2 reversed-phase resin (Thermo Scientific) and kept at 0.5 °C for 3 min, after which the trap column was placed in line with an ACQUITY UPLC BEH C18 1.7 µm 1.0 × 100 mm column (Waters)[83]. Peptides were eluted at 0.5 °C with a gradient of water +0.1% (v/v) formic acid (eluent A) and acetonitrile +0.1% (v/v) formic acid (eluent B) at 60 µL/min flow rate as follows: 0–7 min/95–65% A, 7–8 min/65–15% A, 8–10 min/15% A, guided to a G2-Si HDMS mass spectrometer with ion mobility separation (Waters) and ionized by electrospray ionization (capillary temperature 250 °C, spray voltage 3.0 kV). Mass spectra were acquired over a range of 50–2000 m/z in enhanced high-definition MS (HDMSᴱ)[84,85] or high-definition MS (HDMS) mode for non-deuterated and deuterated samples, respectively. [Glu1]-Fibrinopeptide B standard (Waters) was employed for lock mass correction. During separation of the peptides, the pepsin column was washed three times with 80 µL of 4% (v/v) acetonitrile and 0.5 M guanidine hydrochloride, and blank runs (injection of H₂O) were performed between each sample. Three technical replicates (independent H/D exchange reactions) were measured per incubation time. No correction for HDX back exchange was conducted.

Further data analysis was conducted as described previously[37,38]. Peptides were identified with ProteinLynx Global SERVER 3.0.1 (PLGS, Waters) from the non-deuterated samples acquired with HDMSᴱ by employing low energy, elevated energy, and intensity thresholds of 300, 100, and 1000 counts, respectively. Hereby, the identified ions were matched to peptides with a database containing the amino acid sequences of VirB, porcine pepsin and their reversed sequences with the following search parameters: peptide tolerance = automatic; fragment tolerance = automatic; min fragment ion matches per peptide = 1; min fragment ion matches per protein = 7; min peptide matches per protein = 3; maximum hits to return = 20; maximum protein mass = 250,000; primary digest reagent = non-specific; missed cleavages = 0; false discovery rate = 100. Deuterium incorporation into peptides was quantified with DynamX 3.0 software (Waters). Only peptides that were identified in all non-deuterated samples and with a minimum intensity of 10,000 counts, a maximum length of 40 amino acids, a minimum number of two products, a maximum mass error of 25 ppm, and a retention time tolerance of 0.5 min were considered for analysis. All spectra were manually inspected and, if necessary, peptides were omitted (e.g., in case of low signal-to-noise ratio or presence of overlapping peptides). Mass spectra of VirB samples containing scrambled *virS* were generally of lower quality than the other states and could only be partially assigned.

The observable maximal deuterium uptake of a peptide was calculated by the number of residues minus one (for the N-terminal residue, which after proteolytic cleavage quantitatively loses its deuterium label) minus the number of proline residues contained in the peptide (which lack an exchangeable peptide bond amide proton). For the calculation of HDX in percent, the absolute HDX was divided by the theoretical maximal deuterium uptake and multiplied by 100. The rendering of residue-specific HDX differences from overlapping peptides for any given residue of VirB was performed with DynamX 3.0 by employing the shortest peptide covering any residue. Where multiple

peptides were of the shortest length, the peptide with the residue closest to the C-terminus of the peptide was used.

## In vitro crosslinking

Prior to the crosslinking reactions, VirB or its mutant variants were incubated with 5 mM Tris(2-carboxyethyl)phosphine (TCEP; Sigma, USA) for 1 h at room temperature to fully reduce all cysteine residues. Subsequently, the protein was transferred to reaction buffer (25 mM HEPES/NaOH pH 7.5, 500 mM NaCl, 5 mM MgCl$_2$, 0.1 mM EDTA) using PD SpinTrap G-25 columns (Cytiva, Germany). Crosslinking was performed at room temperature in mixtures containing 10 μM protein, which were supplemented with 1 mM CTP and/or 1 μM of a *virS*-containing double-stranded DNA oligonucleotide (virS-icsB-for/virS-icsB-rev) when appropriate. After the indicated incubation times, bismaleimidoethane (BMOE; dissolved in dimethylsulfoxide) was added to a concentration of 1 mM. After brief mixing and incubation for 1 min, the reaction was stopped by the addition of dithiothreitol-containing SDS sample buffer. As a negative control, samples were treated with dimethylsulfoxide instead of BMOE. The samples were then analyzed by SDS-polyacrylamide gel electrophoresis, and protein was stained with InstantBlue Coomassie Protein Stain (Expedeon, Germany). The gels were imaged with a ChemiDoc MP imaging system (Bio-Rad Laboratories, USA), and the intensities of the different bands were quantified using Image Lab 5.0 software (Bio-Rad Laboratories, USA).

## Mass photometry

The oligomeric states of VirB in the absence and presence of CTP were determined with a TwoMP mass photometer (Refeyn Ltd., UK). Microscope coverslips (1.5 H, 24 × 60 mm, Carl Roth) and CultureWell Reusable Gaskets (CW-50R-1.0, 3 × 1 mm; Grace Bio-labs, USA) were cleaned with three alternating rinsing steps of ddH$_2$O and isopropanol, and dried under a stream of compressed air. The gaskets (six cavities) were applied to the coverslips and mounted on the stage of the mass photometer using immersion oil (Immersol™ 519F; Carl Zeiss, Germany). Prior to each measurement, 18 μL of high-stringency buffer (25 mM HEPES/NaOH pH 7.5, 5 mM MgCl$_2$, 500 mM NaCl and 1 mM DTT), containing 1 mM CTP when required, were pipetted into one of the cavities, and the instrument was focused. Subsequently, two microliters of protein sample were added, resulting in a final protein concentration of ~100 nM. After thorough mixing, the reactions were analyzed for 60 s at 100 frames per second using AcquireMP v2023 R1.1 software (Refeyn Ltd., UK). The instrument was calibrated using a self-made molecular weight standard comprising proteins of known sizes (86 kDa to 430 kDa), and the data obtained were fit to a linear regression model. All data were analyzed using DiscoverMP v2023 R.1.2 (Refeyn Ltd., UK).

## Nucleotide-content analysis

10 μL of VirB (50 μM) were mixed with 40 μL double-distilled water and 150 μL chloroform, followed by 5 s of vigorous shaking, 15 s of heat denaturation at 95 °C and snap-freezing in liquid nitrogen. After thawing and centrifugation (17,300 × g, 10 min, 4 °C), the aqueous phase containing any released nucleotides was withdrawn for HPLC analysis on an Agilent 1260 Infinity system equipped with a Metrosep A Supp 5 – 150/4.0 column (Metrohm, Germany). Samples (10 μL) were injected and eluted at a flow rate of 0.6 mL/min flow rate with 90 mM (NH$_4$)$_2$CO$_3$ pH 9.25. Nucleotides were detected at 260 nm wavelength by comparison of their retention time with those of a commercial standard that was treated as described for the VirB samples.

## Fluorescence microscopy

*E. coli* cells were inoculated to an OD$_{600}$ 0.03 in LB medium and grown for 1 h at 28 °C before the medium was supplemented with 0.1% (w/v) arabinose to induce the production of the indicated mVenus-VirB variants. After further cultivation to an OD$_{600}$ of 0.6, cells were immobilized on 1% (w/v) agarose pads and imaged with a Zeiss Axio Observer.Z1 microscope (Zeiss, Germany) equipped with a Zeiss Plan-Apochromat 100×/1.40 Oil Ph3 M27 objective and a pco.edge 4.2 sCMOS camera (PCO, Germany). An X-Cite 120PC metal halide light source (EXFO, Canada) and ET-YFP filter cubes (Chroma, USA) were used for fluorescence detection. Images were recorded with VisiView 3.3.0.6 (Visitron Systems, Germany) and processed with Fiji 1.53t[86] and Adobe Illustrator CS6 (Adobe Inc., San Jose, USA).

*S. flexneri* cells were grown overnight at 28 °C, diluted to an optical density at 600 nm (OD$_{600}$) of 0.1 in LB medium supplemented with 0.2% (w/v) arabinose and then cultivated for another 2 h at 37 °C. Subsequently, they were immobilized on agarose pads made of 1.5% (w/v) low melting agarose (Sigma–Aldrich) in minimal microscopy medium made of 100 mM HEPES pH 7.2, 5 mM (NH$_4$)$_2$SO$_4$, 100 mM NaCl, 20 mM sodium glutamate, 10 mM MgCl$_2$, 5 mM K$_2$SO$_4$, 50 mM 2-(N-morpholino) ethanesulfonic acid (MES), 50 mM glycine in glass depression slides (Marienfeld) and imaged with a Deltavision Elite Optical Sectioning Microscope equipped with an UPlanSApo 100×/ 1.40 oil objective (Olympus) and a pco.edge 5.5 sCMOS camera (PCO, Germany). Data were recorded with softWoRx 7.0.0 (Applied Precision, Issaquah, WA, USA) and processed with Fiji 1.53t[86] and Adobe Illustrator CS6 (Adobe Inc., San Jose, USA). For each image, a Z-stack containing 11 slices with a spacing of 150 nm was acquired per wavelength and then subjected to maximum intensity projection.

## Immunoblot analysis

Cells were harvested in the exponential growth phase and subjected to immunoblot analysis as described previously[87], using a polyclonal anti-GFP antibody (Sigma, Germany; Cat. #: G1544; RRID: AB_439690) at a 1:10,000 dilution. Goat anti-rabbit immunoglobulin G conjugated with horseradish peroxidase (Revvity, USA; Cat. #: NEF812E001EA; at a 1:20,000 dilution) was used as a secondary antibody. Immunocomplexes were detected with the Western Lightning Plus-ECL chemiluminescence reagent (Perkin Elmer, USA). The signals were recorded with a ChemiDoc MP imaging system (Bio-Rad, Germany) and analyzed using Image Lab software (Bio-Rad, Germany).

## Secretion assay

*S. flexneri* cells were grown overnight at 28 °C, diluted to an OD$_{600}$ of 0.1 in LB medium supplemented with 0.2% (w/v) arabinose and then cultivated at 37 °C for another 2 h. Subsequently, the cultures were supplemented with Congo Red[88] to a final concentration of 200 μg/mL and further incubated at 37 °C for 2–3 h. After the adjustment of all cultures to the same OD$_{600}$, a 2 mL sample of each culture was centrifuged (15,000 × g, 10 min, 4 °C). 1.8 mL of the supernatant were mixed with 0.2 mL of trichloroacetic acid and incubated overnight at 4 °C to precipitate the proteins. Precipitates were collected by centrifugation (15,000 × g, 15 min, 4 °C) and washed two times with ice-cold acetone (15,000 × g, 10 min, 4 °C). The pellet was dried at room temperature, dissolved in SDS sample buffer, incubated at 98 °C for 10 min and subjected to SDS-PAGE. After the staining of proteins with Coomassie Brilliant Blue R-250, the gels were imaged with a ChemiDoc MP imaging system (Bio-Rad Laboratories, USA).

## Shotgun-proteomics analysis

Cells were grown under secretion-inducing conditions as described for the secretion assay. After the adjustment of all cultures to the same OD$_{600}$, 2-mL samples of each culture were centrifuged (15,000 × g, 10 min, 4 °C), and the pellet was resuspended in 100 μL per OD$_{600}$ unit of lysis buffer (2% [w/v] sodium lauroyl sarcosinate (SLS), 100 mM ammonium bicarbonate). The samples were then heated for 10 min at 98 °C and sonicated with a vial tweeter after heating. Proteins were reduced with 5 mM Tris(2-carboxyethyl) phosphine (Thermo Fischer Scientific) at 90 °C for 15 min and alkylated using 10 mM iodoacetamide (Sigma–Aldrich) at 20 °C for 30 min in the dark. Subsequently,

they were precipitated with a sixfold excess of ice-cold acetone and incubation for 2 h at −20 °C, followed by two washing steps with methanol. Dried proteins were reconstituted in 0.2% (w/v) SLS and the amount of total protein was determined using a bicinchoninic acid protein assay (Thermo Scientific). For tryptic digestion, 50 μg of protein was incubated with 1 μg of trypsin (Serva) in 0.5% (w/v) SLS overnight at 30 °C. After digestion, SLS was precipitated by the addition of trifluoroacetic acid (Thermo Fischer Scientific) to a final concentration of 1.5% (v/v). Peptides were desalted using C18 solid-phase extraction cartridges (Macherey-Nagel), which were prepared by the addition of acetonitrile, followed by equilibration with 0.1% (v/v) trifluoroacetic acid. After application to the equilibrated cartridges, the peptides were washed with 5% (v/v) acetonitrile and 0.1% (v/v) trifluoroacetic acid containing buffer and finally eluted with 50% (v/v) acetonitrile and 0.1% (v/v) trifluoroacetic acid.

Dried peptides were reconstituted in 0.1% (v/v) trifluoroacetic acid and then analyzed using liquid-chromatography-mass spectrometry, carried out on an Exploris 480 instrument connected to an Ultimate 3000 RSLC nano and a nanospray flex ion source (all Thermo Scientific). Peptide separation was performed on a reverse phase HPLC column (75 μm × 42 cm) packed in-house with C18 resin (2.4 μm; Dr. Maisch). The following separating gradient was used: 94% solvent A (0.15% [v/v] formic acid in water) and 6% solvent B (0.15% [v/v] formic acid in acetonitrile) to 65% solvent A and 35% solvent B over 60 min at a flow rate of 300 nL/min. Mass spectrometry raw data were acquired on the Exploris 480 instrument in data-independent acquisition (DIA) mode, using a method adopted from previous work[89]. In short, the spray voltage was set to 2.3 kV, the funnel RF level to 40 and the heated capillary temperature to 275 °C, and 445.12003 m/z was used as internal calibrant. For DIA experiments, full MS resolutions were set to 120,000 at m/z 200 and full MS, the AGC (Automatic Gain Control) target was 300% with an IT of 50 ms. The mass range was set to 350–1400. The AGC target value for fragment spectra was set at 3000%. 45 windows of 14 Da were used with an overlap of 1 Da. Resolution was set to 15,000 and IT to 22 ms. Stepped HCD collision energy of 25, 27.5, 30% was used. MS1 data were acquired in profile, MS2 DIA data in centroid mode.

DIA data were analyzed with DIA-NN version 1.8[90] using a Uniprot protein database for *S. flexneri*. Full tryptic digest was allowed with two missed cleavage sites, oxidized methionines and carbamidomethylated cysteins. "Match between runs" and "Remove likely interferences" were enabled. The neural network classifier was set to the single-pass mode, and protein inference was based on genes. The quantification strategy was set to any LC (high accuracy). Cross-run normalization was set to RT-dependent. Library generation was set to smart profiling. DIA-NN outputs were further evaluated using the SafeQuant[91,92] script modified to process DIA-NN outputs. Data were processed with the web tool VolcaNoseR[93] and Adobe Illustrator CS6 (Adobe Inc., USA). The results of the shotgun-proteomics analysis are provided in Supplementary Data 2.

### Protein identification using in-gel digests
Excised gel pieces containing the protein bands of interest were destained in with 60% (v/v) acetonitrile in 20 mM ammonium bicarbonate, further dehydrated using 100% acetonitrile, and rehydrated with 10 mM ammonium bicarbonate. Another round of dehydration was performed, followed by rehydration with 5 mM ammonium bicarbonate and 1 mM TCEP to reduce disulfide bonds at 37 °C for 30 min. After another dehydration step, the gel pieces were rehydrated with 10 mM ammonium bicarbonate and 2 mM iodoacetamide prior to alkylation at 25 °C for 30 min. 10 ng/μL sequencing grade trypsin (Serva) was added, and the digest was allowed to proceed overnight at 30 °C. Subsequently, the supernatant was collected and the gel pieces were dehydrated with 100% acetonitrile to further extract peptides. All peptide extracts from a gel piece were pooled. The peptides were then dried and purified by solid-phase extraction as described above for the shotgun-proteomics analysis, but reducing the duration of the gradient to 30 min. Mass spectrometry raw data were acquired on the abovementioned Exploris 480 instrument in data-dependent acquisition mode with the following settings: MS scan at a resolution of 60,000 full width at half maximum (at m/z 200), followed by MS/MS scans of the most intense ions within 1 s (cycle 1 s). The dynamic exclusion duration was set to 14 s. The ion accumulation time was set to 50 ms (MS) and 50 ms at 17,500 resolution (MS/MS). The automatic gain control (AGC) was set to $3 \times 10^6$ for MS survey scans and $2 \times 10^5$ for MS/MS scans. The quadrupole isolation was 1.5 m/z, collision was induced with an HCD collision energy of 27%. MS raw spectra were searched against a *S. flexneri* protein database using SequestHT within Proteome Discoverer (Thermo Scientific) and evaluated using Scaffold 4 (Proteome Software). For MS searches, the precursor mass tolerance was set to 10 ppm, with 0.02 Da fragment ion mass tolerance and with oxidation (M) and deamidation (N, Q) as variable modifications. Carbamidomethylation (C) was set as fixed modification. Search results were evaluated in Scaffold 4 (Proteome Software, USA).

### Availability of biological material
The plasmids and strains used in this study are available from the corresponding author upon request.

### Reporting summary
Further information on research design is available in the Nature Portfolio Reporting Summary linked to this article.

## Data availability
The mass spectrometry proteomics data generated in this study have been deposited to the ProteomeXchange Consortium via the PRIDE partner repository with the dataset identifier PXD046358. All other data generated in this study are included in the manuscript and the supplementary material. Source data are provided with this paper.

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

## Acknowledgements

We thank Olga Ebers for excellent technical assistance, Jörg Kahnt for support in proteomics sample preparation, Camila Valenzuela Montenegro (Institut Pasteur, France) for providing the *S. flexneri* genome editing protocol and advice on its application, Sven A. Freibert for advice on microscale thermophoresis, Leon Schulte (University of Marburg, Germany) for providing *S. flexneri* strain 2457T, Stephan Gruber for sharing information prior to publication, and Lucas Schnabel for helpful discussions. This work was supported by the University of Marburg (core funding to M.T. and G.B.), the Max Planck Society (core funding to G.K.A.H. and A.D. and Max Planck Fellowships to G.B. and M.T.), the European Research Council (Grant agreement 101097986 – C-SWITCH to M.T.) and the German Research Foundation (DFG) (project 269423233 – TRR 174 to M.T. and project 324652314 – DFG Core Facility for Interactions, Dynamics and Assembly of Biomolecular Structures to G.B.).

## Author contributions

S.J. constructed plasmids, purified proteins, performed biochemical and fluorescence microscopy studies and conducted the secretion assay. W.S. performed the hydrogen-deuterium exchange mass spectrometric analysis. J.H. performed the microscale thermophoresis analysis. J.R. constructed plasmids, purified proteins and conducted fluorescence microscopy studies. K.L. and A.D. generated the *S. flexneri* Δ*virB* mutant. N.S. and S.J. performed the mass photometry analysis. P.I.G. performed the isothermal titration calorimetry analysis. T.G. conducted the whole-proteome analysis and the mass spectrometric identification of proteins from SDS gels. S.J., W.S., J.H., M.O.V., N.S., P.I.G., T.G., A.D., and M.T. analyzed the data. G.K.A.H., G.B., A.D., and M.T. secured funding and supervised the study. M.T. conceived the study. S.J. and M.T. wrote the paper, with input from all other authors.

## Funding

## Competing interests

The authors declare no competing interests.
