## [Peer Review File · Nature Communications]

The virulence regulator VirB from *Shigella flexneri* uses a CTP-dependent switch mechanism to activate gene expressionReviewer #1 (Remarks to the Author):

This is an excellent manuscript by Jakob et al. The team has shown convincingly that VirB from *Shigella flexneri* is a ParB homolog and is a CTP-dependent clamp that recognizes a specific loading site (*virS*) and slides while trapping DNA at the same time. The combination of AlphaFold modeling, biochemistry (bio-layer interferometry, crosslinking, HD-MS experiments), and cell biology all together produced a very convincing set of conclusions. All experiments were performed rigorously and data were presented professionally. There are just two major drawbacks:

1) The manuscript was unable to reveal mechanistically how the sliding clamp can counter the anti-silencing activity of H-NS to activate gene expression. This is the biological activity that makes VirB unique i.e. different from just being another ParB protein. The team speculated a few possibilities on how VirB can activate gene expression in the Discussion section but did not go on to test these; this limits the impact of this manuscript. Because of this, I find the title "The virulence regulator VirB from *Shigella flexneri* uses a CTP-dependent switch mechanism to activate gene expression" and a sentence from the Discussion "Collectively, our work reveals that VirB forms a distinct group of ParB-like proteins that uses a CTP-dependent switch mechanism to associate with target promoter regions and activate gene expression" rather strong regarding linking CTP switch to gene expression activation.

2) It is now established that canonical ParB can bridge and condense DNA, in addition to entrapping and sliding on DNA. Can a CTP-switch VirB also bridge to condense DNA? The team eluded to the supercoiling effect which might activate gene expression in the Discussion section; could the possible bridging activity be the main reason for gene expression activation rather than the sliding activity? I feel this is an important aspect of ParB-like CTP switches that might be worth considering experimentally?

There are some minor comments (in no particular order of importance) for the team to consider:

1) Lines 94-96: "The center of the closed dimer features an opening flanked by non-structured linker regions that is large enough to accommodate a DNA molecule and lined by an abundance of positively charged residues (Figure S1)" This might need quantification. The lumen looks smaller than the equivalents from canonical ParB.

2) Lines 159-161 and Figure 4 A-B: I am slightly surprised that there is no/very little BLI signal in the experiment of VirB with a 215bp DNA fragment in the absence of CTP (and in a high stringency condition). I expect to see at least some signal from VirB binding and staying at *virS* site?

3) Figure 3: the depiction of VirB clamping and binding to a *virS* site by the VBD (in low stringency condition) is rather confusing. In the low stringency condition, if I am correct, the team seems to suggest that DNA binding can be via the VBD or non-specifically via the highly positively-charged CTD or both. The schematic picture in Figure 3A is only suitable for a high-stringency condition. Is there a case to move all data from low-stringency experiments to the supplementary to improve readability? I think data from 500mM salt conditions are the most relevant ones.

4) Lines 160-164 and figure legend 4D: I would caution against calculating that there are 14 dimers of VirB on a 215bp loop from BLI data alone. BLI is not SPR where one can reliably correlate the associate mass to the signal. And 14 dimers x 16-18bp *virS* site = ~224-252bp of an occupied region; but this means that every VirB dimers are right next to each other on a loop, so this is quite unlikely.

5) A control that Q15C did not impact the biological activity of VirB is required; a BLI assay with this mutant should be sufficient.

6) Lines 205-206: "The high efficiency of the crosslinking reaction in the presence of CTP and *virS* could thus indicate that the N-terminal helices are released upon homodimerization of the NBDs." This statement contradicts the HDX-MS results; there was no change in Deuterium for this region (Figure 7).

7) Lines 199-206: the crosslinking at the natural Cys5 is rather strange, but could be interpreted as the evidence for the bridging of two adjacent VirB dimers, rather than the speculation that the N-terminal helix was released. Crosslinking is indeed most likely between 2 opposite subunits of a VirB dimer, but could also be from adjacent VirB dimers.

8) Lines 300-304: I would caution against saying concretely about the dimerization interface is less extensive than those in ParB dimers based on an AlphaFold model alone.

9) Maybe VirB has some low CTPase activity (it does not make much biological sense for VirB to lack the CTPase activity completely). The BLI data and crosslinking data when CTPyS is used instead would be revealing.

Reviewer #2 (Remarks to the Author):

The authors have used a variety of methods to explore the properties of the VirB transcriptional anti-silencing protein from *Shigella flexneri*, whose function, derepression of the *S. flexneri* virulence regulon, has been known for several decades. Their work advances our knowledge of the molecular mechanism by which the ParB-like VirB protein remodels H-NS nucleoprotein complexes at VirB target promoters, leading to their derepression.

We learn that VirB binds CTP, a property that is shared with other ParB-like proteins, but does not hydrolyse it, even in the presence of VirB's DNA binding site, *virS*. Under stringent (i.e. high salt) conditions, VirB displays *virS* binding specificity in the presence of CTP. Evidence from some nicely designed experiments supports the authors' proposal that VirB acts as a sliding clamp on DNA, that clamp closure occurs optimally in the presence of CTP and *virS*, with the latter behaviour involving CTP+*virS* stimulation of dimer formation at the VirB N-terminus. Some in vivo studies with two VirB derivatives deficient in CTP binding show that this binding activity is a requirement for the loading of the protein onto *virS* in bacterial cells.

The authors focus on VirB interaction with the *virS* sequence that is upstream of the *icsB* gene within the *Shigella* large virulence plasmid's Entry Region. The *ipgD* virulence gene lies upstream of *icsB* and is transcribed from the opposite DNA strand. Like *icsB*, *ipgD* is VirB-dependent for transcription anti-silencing. Is the sliding of VirB on DNA a bidirectional process, and if so, do *icsB* and *ipgD* share the same *virS* sequence? As this *virS* element (5'-TTTCATcATGAAA-3') is 94 bp upstream of the *ipgD* transcription start site, and 112 bp upstream of the *icsB* transcription start site (DOI: 10.1128/jb.179.21.6537-6550.1997), this seems plausible. Would this make physiological sense in the timing of the assembly of the *Shigella* type III secretion system and the production of the effector proteins that must pass through it?

The sliding of ParB CTP-binding proteins can be blocked by other DNA binding proteins (doi: 10.1126/science.283.5401.546; doi: 10.1126/science.aay3965; doi: 10.7554/eLife.53515.). Might interference of this type modulate the positive effects of VirB on its target genes? (The Lac repressor/operator complex is an effective roadblock to VirB movement at the *icsP* promoter: DOI: 10.1111/mmi.13932 reference 49 in the present study).

Lines 58-61, *virS*, VirB and DNA supercoiling. The work to which the authors refer (Reference 53, Picker et al., 2023) was performed with the *icsP* gene, a different gene in a different location on the virulence plasmid to the *icsB*-*ipgD*-associated *virS* element that is the focus of the current study. The distinction is important because, in keeping with the twin-domain supercoiling model of Liu and Wang (DOI: 10.1073/pnas.84.20.7024), the back-to-back promoters of the divergently transcribed *icsB* and *ipgD* genes would be expected to pump negative supercoils into the DNA between them – the region where the *virS* element is almost centrally located. This may have implications for the sustained transcription of these genes in the presence of the abundant H-NS silencing protein.

In light of the findings of Picker et al. at *icsP* (Ref 53), and the likely generation by divergent transcription units of a locally confined domain of negatively supercoiled DNA with *virS* at its center, do the authors consider it likely that *VirB* co-operates with topoisomerase I to remodel the H-NS-DNA transcription-silencing complex in the *icsB-ipgD* intergenic region? Previous work showing that loss of topoisomerase I production in *S. flexneri* down-regulates transcription in the *VirB*-dependent virulence regulon suggests that this is likely (DOI: 10.1111/j.1365-2958.1993.tb01127.x).

Line 22 "*VirB*, which activates the expression of ~50 genes," While this is a large set of genes, perhaps mention that these genes are organised in a small number of large operons, so the number of binding sites that has to be targeted by *VirB* is correspondingly small.

Line 60-61: "However, the underlying mechanism remains unclear." Which aspects of the underlying mechanism remain unclear? By spelling out what it is that remains unclear, you will help the reader to understand the rationale for the experiments that you propose to perform and you will assist the reader in assessing the extent to which you succeed in reaching your goal.

Line 129: "under any of the condition tested" should read "under any of the conditions tested"

Line 172: "we analysed for the interaction" should read: "we analysed the interaction"

Line 354: "induces the formation of positive supercoils". How might *VirB* achieve a change in DNA linking number without breaking and religating DNA strands? Does *VirB* simply displace negative supercoils, leading to a compensatory relaxation of these negative supercoils by topoisomerase I? When this relaxed DNA is then isolated (free of protein) and studied by electrophoresis in the presence of an intercalating agent, it will adopt a positively supercoiled conformation. This is not equivalent to the formation of positive supercoils *in vivo*. It might be safer to state that *VirB* interaction with DNA leads to a loss of negative supercoils *in vivo* (with the relaxed DNA being converted to a positively supercoiled state by chloroquine intercalation during gel electrophoresis).

Table S2 (Oligonucleotides): the inverted repeat part of *virS* from the *icsB* gene is shown in bold on several lines in the table and is reproduced here, in upper case: 5'-GTTTCATcaTGAAAt-3'. The first 'G' residue in the version shown in Table S2 does not have a complementary counterpart in the region of dyad symmetry. In addition, two bases that are mutually complementary in the central region are not highlighted. A strict application of a dyad symmetry requirement would produce the following: 5'-gTTTCATcATGAAAt-3'.

The *virS* sequence shown in Fig 4 (panel A) appears to have been written in its inverted orientation: 5'-ATTTCAtgaTGAAA-3'. Was this done intentionally?

Reviewer #3 (Remarks to the Author):

The article referenced NCOMMS-23-24362 by Jakob et al reports that *VirB*, a transcription regulator involved in bacterial pathogenesis, functions as a CTP dependent molecular switch. In a similar manner to *ParB*, *VirB* binds to CTP, clamps onto DNA at specific locations, and subsequently moves along the DNA strand. Mutations that disrupt CTP binding have a negative effect on the closure of the *VirB* clamp and impede its function both *in vivo* and *in vitro*. Overall, the article is well-written and presents a variety of interesting findings that are generally convincing. However, there are certain points that require to be addressed.

1. Validation of the structure model

The authors constructed a structural model of the *VirB* dimer; however, they did not provide any validation for this model. To establish confidence in the proposed interactions, it is recommended to incorporate metrics such as the predicted interface TM score (ipTM) or the predicted DockQ value (pDockQ) for the dimeric model. Additionally, it is worth exploring the possibility of other

oligomeric states, such as a trimer or tetramer, by constructing and comparing confidence levels of models representing these conformations. Furthermore, understanding the oligomeric states of VirB in solution is of significance and should be addressed in the study.

2. CTP dependent closure of the VirB clamp

-To provide a more accurate evaluation of the relationship between VirB and ParB proteins, it is necessary to present structural comparison results for individual domains, such as VBD and CTD, between VirB and chromosomally encoded ParB proteins as well.

-To demonstrate the reliability of the obtained binding affinity, it is suggested to depict the raw data fitting curves for ITC in Figure 2A. Additionally, providing the R square and N values for each titration is recommended.

-To strengthen their conclusion regarding the requirement of CTP for the closure of the VirB dimer, it is suggested that the authors conduct additional BMOE cross-linking experiments. Specifically, they could investigate the R93A and R94A variants of VirB, which are unable to bind CTP.

-In lines 192-193, the authors stated that during the process of ring closure, the newly incorporated C15 residues in the two NBDs of VirB are positioned adjacently to each other. To support this claim, similar to Figure S6, the localization of Q15C in the dimeric model of VirB should be labeled, with the distance between the two Q15C residues indicated.

-Under the assumption that stringent conditions can effectively prevent non-specific DNA binding, it is puzzling why short DNA sequences with a central virS site fail to stably associate with VirB (Figure 5B), whereas longer DNA fragments containing the virS site can form stable associations with VirB (Figure 4). The authors should offer an explanation to elucidate this phenomenon.

3. The working model

The authors assert that VirB employs a loading mechanism dependent on CTP at virS site and subsequently undergoes lateral movement on DNA following clamp closure. However, they cannot rule out an alternative mechanism involving virS-VirB interaction and nucleation when examining the binding of VirB to DNA fragments adjacent to the virS site, besides the lateral sliding mechanism.

We thank the three reviewers for their constructive criticism, which helped to significantly improve our manuscript. Please see below for our response to the different points raised.

REVIEWER COMMENTS

Reviewer #1

This is an excellent manuscript by Jakob et al. The team has shown convincingly that VirB from *Shigella flexneri* is a ParB homolog and is a CTP-dependent clamp that recognizes a specific loading site (*virS*) and slides while trapping DNA at the same time. The combination of AlphaFold modeling, biochemistry (bio-layer interferometry, crosslinking, HD-MS experiments), and cell biology all together produced a very convincing set of conclusions. All experiments were performed rigorously and data were presented professionally. There are just two major drawbacks:

1) The manuscript was unable to reveal mechanistically how the sliding clamp can counter the anti-silencing activity of H-NS to activate gene expression. This is the biological activity that makes VirB unique i.e. different from just being another ParB protein. The team speculated a few possibilities on how VirB can activate gene expression in the Discussion section but did not go on to test these; this limits the impact of this manuscript. Because of this, I find the title “The virulence regulator VirB from *Shigella flexneri* uses a CTP-dependent switch mechanism to activate gene expression” and a sentence from the Discussion “Collectively, our work reveals that VirB forms a distinct group of ParB-like proteins that uses a CTP-dependent switch mechanism to associate with target promoter regions and activate gene expression” rather strong regarding linking CTP switch to gene expression activation.

The clarification of the precise mechanism underlying the silencing activity of VirB is not a straightforward task and thus beyond the scope of the present study. However, we believe that the demonstration of the loading-and-sliding activity of ParB and the establishment of experimental approaches to investigate this activity *in vitro* are a significant step forward in the understanding of VirB-mediated gene regulation and set the basis for such future studies.

We agree that the previous version of our manuscript did not provide any direct evidence that VirB requires CTP binding to regulate virulence gene expression in *S. flexneri*, although the data obtained strongly supported this hypothesis. We have therefore set out to investigate the functionality of the CTP-binding-defective VirB variants in *S. flexneri* cells. Localization studies show that wild-type VirB forms one or multiple foci in *S. flexneri*, whereas the R93A and R94A variants have a diffuse localization, indicating that they are no longer loaded onto promoter regions on the pINV virulence plasmid. Moreover, total proteome analyses demonstrate that virulence gene expression is unaffected in a newly constructed *S. flexneri* $\Delta virB$ mutant expressing wild-type *virB* from a replicating plasmid but completely abolished in mutant cells producing the R93A or R94A variants. Consistent with these results, we observe that mutant cells producing either of the mutant variants no longer secrete effectors to the growth medium. Together, these results provide firm evidence that the CTP-dependent switch mechanism of VirB is critical for virulence gene expression in *S. flexneri* and, thus, verify the statements highlighted above by Reviewer #1.

2) It is now established that canonical ParB can bridge and condense DNA, in addition to entrapping and sliding on DNA. Can a CTP-switch VirB also bridge to condense DNA? The team eluded to the supercoiling effect which might activate gene expression in the Discussion section; could the possible bridging activity be the main reason for gene expression activation rather than the sliding activity? I feel this is an important aspect of ParB-like CTP switches that might be worth considering experimentally?

The bridging activity of ParB has so far only been shown for a single (*B. subtilis*) ParB homolog *in vitro*, and its *in vivo* relevance remains unclear. However, it is possible that bridging is a common property of ParB proteins that may also apply to VirB and contribute to its anti-silencing effect. We have now included this possibility in the Discussion.

Since bridging interactions between distal VirB clamps are not possible without previous sliding, the lateral sliding of VirB away from *virS* sites is, by necessity, a critical factor in the underlying mechanism. In general, it is very difficult to test the potential relevance of bridging in the function of VirB. Even for *B. subtilis* ParB, the precise mode of interaction between two bridging ParB clamps is unclear, although the data obtained suggest that it may involve face-to-face interactions between the ParB/Srx domains of the two interacting ParB dimers. Therefore, it has so far been impossible to specifically block these interactions in order to disentangle the contributions of sliding and bridging in the function of ParB *in vitro* and *in vivo*. Notably, the formation of bridges by *B. subtilis* ParB was shown to require CTP hydrolysis, suggesting that it relies on the transient opening of ParB clamps for subsequent homodimerization of the ParB/Srx domains *in trans*. Given the extremely low CTPase activity of VirB, these opening events may be relatively rare, thus potentially making abundant bridge formation unlikely. Given these issues, it is currently not possible to address the potential role of bridging in the anti-silencing activity of VirB.

There are some minor comments (in no particular order of importance) for the team to consider:

1) Lines 94-96: “The center of the closed dimer features an opening flanked by non-structured linker regions that is large enough to accommodate a DNA molecule and lined by an abundance of positively charged residues (Figure S1)” This might need quantification. The lumen looks smaller than the equivalents from canonical ParB.

We have now included a model of the VirB dimer with a double-stranded DNA oligonucleotide occupying the lumen formed by the two non-structured linker regions (new Supplementary Figure 2b). This model verifies that there is sufficient space to accommodate a DNA molecule in this region in a closed VirB dimer.

2) Lines 159-161 and Figure 4 A-B: I am slightly surprised that there is no/very little BLI signal in the experiment of VirB with a 215bp DNA fragment in the absence of CTP (and in a high stringency condition). I expect to see at least some signal from VirB binding and staying at *virS* site?

The absence of a clear binding signal at the concentrations of VirB used indicates that the interaction of VirB with *virS* sites is very transient in high-stringency buffer, in agreement with the absence of any obvious change in the HDX pattern of VirB under these conditions. This reduced affinity may be explained by the disruption of electrostatic interactions between VirB and the DNA phosphate backbone, which also contribute to the overall affinity of VirB to *virS* sites. However, apparently, transient interactions with *virS* appear to be sufficient to trigger the closure of the VirB clamps and thus promote their loading onto DNA

in the presence of CTP. We have now included in the presentation of the HDX data a cross-reference to the BLI data to make this point clearer.

3) Figure 3: the depiction of VirB clamping and binding to a *virS* site by the VBD (in low stringency condition) is rather confusing. In the low stringency condition, if I am correct, the team seems to suggest that DNA binding can be via the VBD or non-specifically via the highly positively-charged CTD or both. The schematic picture in Figure 3A is only suitable for a high-stringency condition. Is there a case to move all data from low-stringency experiments to the supplementary to improve readability? I think data from 500mM salt conditions are the most relevant ones.

There is no reason to believe that VirB cannot be loaded onto DNA through *virS* binding and subsequent sliding in low-stringency conditions. However, the specific loading process is likely to be masked by excessive non-specific binding through the VBD and CTD. To make this point clear we would prefer to keep the schematic as it is.

Previous *in vitro* studies of the DNA-binding behavior of VirB has, in most part, been performed with buffers that contained even less salt than the low-stringency buffer used in our study. To demonstrate the high non-specific DNA binding activity of VirB., highlight the problems it poses on the analysis of VirB-DNA interactions and thus facilitate the interpretation of previous results, we would like to keep this part of our results in the main body of the paper.

4) Lines 160-164 and figure legend 4D: I would caution against calculating that there are 14 dimers of VirB on a 215bp loop from BLI data alone. BLI is not SPR where one can reliably correlate the associate mass to the signal. And 14 dimers x 16-18bp *virS* site = ~224-252bp of an occupied region; but this means that every VirB dimers are right next to each other on a loop, so this is quite unlikely.

We have now removed this statement from the paper.

5) A control that Q15C did not impact the biological activity of VirB is required; a BLI assay with this mutant should be sufficient.

We have performed the requested control experiment (new Supplementary Figure 8). The results show that the mutant protein is fully functional *in vitro*.

6) Lines 205-206: "The high efficiency of the crosslinking reaction in the presence of CTP and *virS* could thus indicate that the N-terminal helices are released upon homodimerization of the NBDs." This statement contradicts the HDX-MS results; there was no change in Deuterium for this region (Figure 7).

We agree that this statement is not supported by the HDX data and have thus removed this sentence from the paper.

7) Lines 199-206: the crosslinking at the natural Cys5 is rather strange, but could be interpreted as the evidence for the bridging of two adjacent VirB dimers, rather than the speculation that the N-terminal helix was released. Crosslinking is indeed most likely between 2 opposite subunits of a VirB dimer, but could also be from adjacent VirB dimers.

As mentioned above, bridges between two distal, transiently opened ParB dimers were postulated to form through face-to-face interactions between the ParB/Srx domains of these dimers. If this is true, the mode of interaction between the ParB/Srx domains would be identical in bridging as compared to sliding ParB clamps. Assuming, that potential bridging interactions between VirB clamps are based on a similar mechanism, it is unlikely that the crosslinking of wild-type VirB is based on the establishment of bridging interactions. Moreover, the DNA molecules used in the crosslinking assays are short double-stranded oligonucleotides, which greatly reduces the chance that transiently opened VirB clamps meet in solution before they close again. We believe that the high efficiency of crosslinking may indicate that the N-terminal helix is simply not firmly attached to the remaining part of the protein but in a dynamic equilibrium between a bound and a free state. We have now added this possibility as an explanation in the presentation of the results.

8) Lines 300-304: I would caution against saying concretely about the dimerization interface is less extensive than those in ParB dimers based on an AlphaFold model alone.

This statement has now been removed from the paper.

9) Maybe VirB has some low CTPase activity (it does not make much biological sense for VirB to lack the CTPase activity completely). The BLI data and crosslinking data when CTPyS is used instead would be revealing.

Given the relatively high scattering of the data obtained for the three replicates in the first version of our manuscript, we have repeated the measurement and performed three technical replicates for each of the biological replicates to improve the precision of our measurements. The new results show that VirB has a very weak basal CTPase activity, which increases twofold in the presence of *virS*-containing DNA. Nevertheless, the maximal turnover number obtained is still more than tenfold lower than that measured for *M. xanthus* ParB. It remains to be determined whether the CTPase activity of VirB is relevant to its function *in vivo*. We ran out of CTPyS, which is only available through custom synthesis, and did not manage to obtain new material within the timeframe of the revision process. Therefore, we were unfortunately unable to perform the requested experiments.

Reviewer #2

The authors have used a variety of methods to explore the properties of the VirB transcriptional anti-silencing protein from *Shigella flexneri*, whose function, derepression of the *S. flexneri* virulence regulon, has been known for several decades. Their work advances our knowledge of the molecular mechanism by which the ParB-like VirB protein remodels H-NS nucleoprotein complexes at VirB target promoters, leading to their derepression.

We learn that VirB binds CTP, a property that is shared with other ParB-like proteins, but does not hydrolyse it, even in the presence of VirB's DNA binding site, *virS*. Under stringent (i.e. high salt) conditions, VirB displays *virS* binding specificity in the presence of CTP. Evidence from some nicely designed experiments supports the authors' proposal that VirB acts as a sliding clamp on DNA, that clamp

closure occurs optimally in the presence of CTP and *virS*, with the latter behaviour involving CTP+*virS* stimulation of dimer formation at the VirB N-terminus. Some in vivo studies with two VirB derivatives deficient in CTP binding show that this binding activity is a requirement for the loading of the protein onto *virS* in bacterial cells.

The authors focus on VirB interaction with the *virS* sequence that is upstream of the *icsB* gene within the Shigella large virulence plasmid's Entry Region. The *ipgD* virulence gene lies upstream of *icsB* and is transcribed from the opposite DNA strand. Like *icsB*, *ipgD* is VirB-dependent for transcription anti-silencing. Is the sliding of VirB on DNA a bidirectional process, and if so, do *icsB* and *ipgD* share the same *virS* sequence? As this *virS* element (5'-TTTCATcATGAAA-3') is 94 bp upstream of the *ipgD* transcription start site, and 112 bp upstream of the *icsB* transcription start site (DOI: 10.1128/jb.179.21.6537-6550.1997), this seems plausible. Would this make physiological sense in the timing of the assembly of the Shigella type III secretion system and the production of the effector proteins that must pass through it?

The sliding of VirB clamps on DNA is likely to be a random process and thus bidirectional. Consistent with this notion, the single *virS* site in the *icsB-ipgD* intergenic region is sufficient to regulate the expression of both operons in the Entry Region. This previously established fact is now underscored by the results shown in the new Figure 9. We have also included a schematic showing representative target promoter regions of VirB (new Supplementary Figure 4) and now state that we “amplified the *icsB-ipgD* intergenic region of pINV with its centrally located *virS* site, which mediates the VirB-dependent regulation of two divergent large virulence operons in the so-called Entry Region of *S. flexneri*”. However, for simplicity we prefer to refer to this *virS* site as the *virS* site in the *icsB* promoter region.

It makes sense that the formation of the T3SS is coordinated with the production of secreted effectors.

The sliding of ParB CTP-binding proteins can be blocked by other DNA binding proteins (doi: 10.1126/science.283.5401.546; doi: 10.1126/science.aay3965; doi: 10.7554/eLife.53515.). Might interference of this type modulate the positive effects of VirB on its target genes? (The Lac repressor/operator complex is an effective roadblock to VirB movement at the *icsP* promoter: DOI: 10.1111/mmi.13932 reference 49 in the present study).

This is an interesting aspect, which we have now included in the discussion.

Lines 58-61, *virS*, VirB and DNA supercoiling. The work to which the authors refer (Reference 53, Picker et al., 2023) was performed with the *icsP* gene, a different gene in a different location on the virulence plasmid to the *icsB-ipgD*-associated *virS* element that is the focus of the current study. The distinction is important because, in keeping with the twin-domain supercoiling model of Liu and Wang (DOI: 10.1073/pnas.84.20.7024), the back-to-back promoters of the divergently transcribed *icsB* and *ipgD* genes would be expected to pump negative supercoils into the DNA between them – the region where the *virS* element is almost centrally located. This may have implications for the sustained transcription of these genes in the presence of the abundant H-NS silencing protein.

Given that the supercoiling state of *virS*-containing promoter regions appears to be important for VirB-mediated anti-silencing, the directionality of transcription of the flanking genes is likely to have an effect on the regulatory output. The divergent *icsB* and *ipgD* promoters will cause increased negative supercoiling in the intergenic region they share, because they not only lead to the unwinding of DNA from both sides but also hinder the diffusion of negative supercoils away from the promoter region. As suggested by Reviewer #2 below, topoisomerase I may be required for efficient VirB-mediated anti-silencing in this case. These aspects are now mentioned in the Discussion.

In light of the findings of Picker et al. at *icsP* (Ref 53), and the likely generation by divergent transcription units of a locally confined domain of negatively supercoiled DNA with *virS* at its center, do the authors consider it likely that VirB co-operates with topoisomerase I to remodel the H-NS-DNA transcription-silencing complex in the *icsB*-*ipgD* intergenic region? Previous work showing that loss of topoisomerase I production in *S. flexneri* down-regulates transcription in the VirB-dependent virulence regulon suggests that this is likely (DOI: 10.1111/j.1365-2958.1993.tb01127.x).

We agree that topoisomerase I is likely to facilitate the anti-silencing activity of VirB by removing excess negative supercoiling in the promoter regions of actively transcribed genes. We have now included this point in the Discussion.

Line 22 "VirB, which activates the expression of ~50 genes," While this is a large set of genes, perhaps mention that these genes are organised in a small number of large operons, so the number of binding sites that has to be targeted by VirB is correspondingly small.

We have now modified the introduction as follows: "... which activates the expression of ~50 genes, *most of which are organized into two large operons* coding for the structural components of the T3SS and for effectors mediating host invasion."

Line 60-61: "However, the underlying mechanism remains unclear." Which aspects of the underlying mechanism remain unclear? By spelling out what it is that remains unclear, you will help the reader to understand the rationale for the experiments that you propose to perform and you will assist the reader in assessing the extent to which you succeed in reaching your goal.

We have now specified our statement as follows: "However, the molecular mechanism underlying the spreading of VirB and its effect on DNA topology are incompletely understood."

Line 129: "under any of the condition tested" should read "under any of the conditions tested"

We have re-written the section containing indicated typographic error.

Line 172: "we analysed for the interaction" should read: "we analysed the interaction"

Corrected.

Line 354: "induces the formation of positive supercoils". How might VirB achieve a change in DNA linking

number without breaking and religating DNA strands? Does VirB simply displace negative supercoils, leading to a compensatory relaxation of these negative supercoils by topoisomerase I? When this relaxed DNA is then isolated (free of protein) and studied by electrophoresis in the presence of an intercalating agent, it will adopt a positively supercoiled conformation. This is not equivalent to the formation of positive supercoils *in vivo*. It might be safer to state that VirB interaction with DNA leads to a loss of negative supercoils *in vivo* (with the relaxed DNA being converted to a positively supercoiled state by chloroquine intercalation during gel electrophoresis).

Our previous statement was indeed misleading because VirB loading does not generate a positively supercoiled DNA topology but reduces the degree of negative supercoiling. We have now changed the wording accordingly. It is difficult to imagine how VirB binding could actively remodel DNA and displace negative supercoils, assuming that its movement is driven only by random diffusion. As mentioned in the Discussion, we think that VirB clamps could potentially introduce transient changes in the supercoiling state of target promoter regions by distorting the DNA upon ring closure. However, future work is required to fully clarify the precise anti-silencing mechanism.

Table S2 (Oligonucleotides): the inverted repeat part of *virS* from the *icsB* gene is shown in bold on several lines in the table and is reproduced here, in upper case: 5'-GTTTCAtcaTGAAAt-3'. The first 'G' residue in the version shown in Table S2 does not have a complementary counterpart in the region of dyad symmetry. In addition, two bases that are mutually complementary in the central region are not highlighted. A strict application of a dyad symmetry requirement would produce the following: 5'-gTTTCATcATGAAAt-3'.

We have now indicated the dyad symmetry of the binding sites properly in the oligonucleotides used. We chose to only highlight the most highly conserved bases and not the more highly variable intervening part, although the nature of the intervening bases has been shown to have an effect on the efficiency of VirB binding.

The *virS* sequence shown in Fig 4 (panel A) appears to have been written in its inverted orientation: 5'-ATTTCAtgaTGAAA-3'. Was this done intentionally?

Initially, we chose to use the inverse orientation because we referred to it as the *virS* site in the *icsB* promoter region. However, for consistency and since we now refer to it as the *virS* site in the *icsB-*ipgD** intergenic region, we now use the normal orientation, as given the schematic in Supplementary Figure 4.

Reviewer #3

The article referenced NCOMMS-23-24362 by Jakob et al reports that VirB, a transcription regulator involved in bacterial pathogenesis, functions as a CTP dependent molecular switch. In a similar manner to ParB, VirB binds to CTP, clamps onto DNA at specific locations, and subsequently moves along the DNA strand. Mutations that disrupt CTP binding have a negative effect on the closure of the VirB clamp and impede its function both *in vivo* and *in vitro*. Overall, the article is well-written and presents a variety of

interesting findings that are generally convincing. However, there are certain points that require to be addressed.

1. Validation of the structure model

The authors constructed a structural model of the VirB dimer; however, they did not provide any validation for this model. To establish confidence in the proposed interactions, it is recommended to incorporate metrics such as the predicted interface TM score (ipTM) or the predicted DockQ value (pDockQ) for the dimeric model. Additionally, it is worth exploring the possibility of other oligomeric states, such as a trimer or tetramer, by constructing and comparing confidence levels of models representing these conformations. Furthermore, understanding the oligomeric states of VirB in solution is of significance and should be addressed in the study.

We now give the ipTM score (0.776) for the dimeric complex in the legend to Figure 1b.

Moreover, we generated models of tetrameric VirB. However, the predictions consistently gave highly intertwined aggregates that do not appear to be biologically relevant:

Top-ranking model of a VirB tetramer generated by AlphaFold-Multimer.

To clarify the oligomeric state of VirB in solution, we performed mass photometry analyses of VirB in high-stringency conditions in the absence and presence of CTP. In both conditions, VirB is exclusively detected as a dimer (new Supplementary Figure 1).

2. CTP dependent closure of the VirB clamp

- To provide a more accurate evaluation of the relationship between VirB and ParB proteins, it is necessary to present structural comparison results for individual domains, such as VBD and CTD, between VirB and chromosomally encoded ParB proteins as well.

- To demonstrate the reliability of the obtained binding affinity, it is suggested to depict the raw data fitting curves for ITC in Figure 2A. Additionally, providing the R square and N values for each titration is recommended.

The fitting curves and the N values are now shown in the new Supplementary Figure 3a. Our software does not provide an R^2 value, because in the strict sense the coefficient of determination only applies to linear fits. However, the graphs show that the fitted curve describes the measured data very well.

- To strengthen their conclusion regarding the requirement of CTP for the closure of the VirB dimer, it is suggested that the authors conduct additional BMOE cross-linking experiments. Specifically, they could investigate the R93A and R94A variants of VirB, which are unable to bind CTP.

We have now performed crosslinking studies of the R93A and R94A variants. The results show that both of them fail to transition to the closed state in the presence of CTP (new Supplementary Figure 16).

- In lines 192-193, the authors stated that during the process of ring closure, the newly incorporated C15 residues in the two NBDs of VirB are positioned adjacently to each other. To support this claim, similar to Figure S6, the localization of Q15C in the dimeric model of VirB should be labeled, with the distance between the two Q15C residues indicated.

We have now included a model of VirB showing the positions of the newly introduced C15 residues (new Figure 6a) and give the distance (6.1 Å) in the legend.

- Under the assumption that stringent conditions can effectively prevent non-specific DNA binding, it is puzzling why short DNA sequences with a central *virS* site fail to stably associate with VirB (Figure 5B), whereas longer DNA fragments containing the *virS* site can form stable associations with VirB (Figure 4). The authors should offer an explanation to elucidate this phenomenon.

The key difference between the experiments in Figures 4 and 5 is the nature of the DNA ends. In Figure 4, both ends of the DNA molecules are immobilized on the BLI sensor surface, so that VirB clamps are trapped on the DNA after their release from *virS*. In Figure 5, by contrast, the DNA molecules are only attached at one end. As a consequence, they slide off the DNA once they transition to the closed-clamp state.

3. The working model

The authors assert that VirB employs a loading mechanism dependent on CTP at *virS* site and subsequently undergoes lateral movement on DNA following clamp closure. However, they cannot rule out an alternative mechanism involving *virS*-VirB interaction and nucleation when examining the binding of VirB to DNA fragments adjacent to the *virS* site, besides the lateral sliding mechanism.

The data presented in Figure 5 strongly suggest that VirB molecules start to slide laterally on the DNA after their loading at *virS*. If the accumulation of VirB on the DNA molecules were mediated by interactions between VirB dimers that are stably associated with *virS* and distal VirB dimers that are bound non-specifically to DNA, then the level of the BLI signal should be largely independent of the way in which DNA is immobilized on the BLI sensors. Moreover, our results show that the non-specific DNA-binding activity of VirB is completely abolished in high-stringency conditions, which makes it unlikely that VirB dimers can be loaded efficiently without previous interaction with *virS*.

Reviewer #1 (Remarks to the Author):

The authors have addressed all concerns, I have no further comments. Thank you.

Reviewer #3 (Remarks to the Author):

The authors have addressed all important concerns raised. I have no further comments to add and consequently recommend the acceptance of the paper.